# GRADIENT-BASED INFERENCE OF TASK ABSTRACTIONS FOR GENERALIZATION IN NEURAL NETWORKS

## ABSTRACT

Humans and many animals show remarkably adaptive behavior and can respond differently to the same input depending on their internal goals. The brain not only represents the intermediate abstractions needed to perform a computation but also actively maintains a representation of the computation itself (task abstraction). Such separation of the computation and its abstraction is associated with faster learning, flexible decision-making, and broad generalization capacity. We investigate if such benefits might extend to neural networks trained with task abstractions. For such benefits to emerge, one needs a task inference mechanism that possesses two crucial abilities: First, the ability to infer abstract task representations when no longer explicitly provided (task inference), and second, manipulate task representations to adapt to novel problems (task recomposition). To tackle this, we cast task inference as an optimization problem from a variational inference perspective and ground our approach in an expectation-maximization framework. We show that gradients backpropagated through a neural network to a task representation layer are an efficient heuristic to infer current task demands, a process we refer to as gradient-based inference (GBI). Further iterative optimization of the task representation layer allows for recomposing abstractions to adapt to novel situations. Using a toy example, a novel image classifier, and a language model, we demonstrate that GBI provides higher learning efficiency and generalization to novel tasks and limits forgetting. Moreover, we show that GBI has unique advantages such as preserving information for uncertainty estimation and detecting out-of-distribution samples.

## 1 INTRODUCTION

Cognitive science and neuroscience hold a prominent place for (top-down) abstract task representations in many accounts of advanced cognitive functions in animals, including humans (Niv, 2019). The brain not only represents the intermediate abstractions needed to perform a task but also actively maintains a representation of the task itself (i.e., task abstraction, (Mante et al., 2013; Rikhye et al., 2018; Zhou et al., 2019; Vaidya et al., 2021; Hummos et al., 2022)). Such separation of the computation and its abstraction is theorized to support learning efficiency, as well as adaptive behavior. First, regarding learning efficiency, task abstractions facilitate learning by organizing experiences collected (Yu et al., 2021). Human participants learn faster once they discover the underlying task structure (Badre et al., 2010; Collins, 2017; Vaidya et al., 2021; Castañón et al., 2021). In fact, humans have a bias to discover and use such latent task structures even when they do not exist or might hurt their performance (Gaissmaier & Schooler, 2008; Collins, 2017). Second, regarding adaptive behavior, previous work has shown that the brain updates these task abstractions to meet current task demands of the environment and (re)composes them to solve new problems (Miller & Cohen, 2001; Collins & Koechlin, 2012). Indeed, depending on the context, the brain is able to respond to the same sensory input in novel ways by composing an appropriate task abstraction that guides processing of the input (Rikhye et al., 2018; Tafazoli et al., 2024), supporting flexible decision making and generalization to novel situations (Collins & Koechlin, 2012; Vaidya et al., 2021).

Traditional artificial neural networks (ANNs) architectures have a static computational graph that entangles input processing with task inference. For task-dependent computations in ANNs, one popular solution is to build task representations into the structure of the network by using modular networks; where each module could represent a given task (Andreas et al., 2016; Kirsch et al., 2018; Goyal et al., 2019). Alternatively, one could regularize the weight updates such as to maximize an

orthogonal computational space for each task (Kirkpatrick et al., 2017; Masse et al., 2018). Our framework is most related to models that add task encoding input layers, which provide information about the current task demands to the network (Yang et al., 2019; Hurtado et al., 2021; Wang & Zhang, 2022; Kumar et al., 2022; 2024).

However, these previous models lack two important task inference features which we unpack in what follows and motivate our work. First, these models do not propose a mechanism to efficiently identify these tasks if they appear again in data, and thereby flexibly (re)adapt to the demands of previously encountered task. Second, and perhaps most importantly, these models lack a principled way of recomposing previously learned task abstractions, thereby minimizing the necessity for new learning (i.e. parameters update) to adapt to new situations (Lake et al., 2017). Importantly, neural systems impose a unique constraint on such task inference and recomposition mechanisms. As the neural system 'learns' through updating parameters, previous values in the task abstractions layer might become outdated. Thereby, the task inference mechanisms must be able to adjust dynamically to changes in neural parameters.

To address the challenge of an efficient inference mechanism that can capture the two properties just described, and maintain alignment to neural parameters, we propose Gradient-Based Inference (GBI) of task abstractions. This solution is grounded in variational inference, offering a robust and flexible framework. For an intuitive example of this framework, we consider that during upbringing, we may observe situations where people's feelings were labeled as sadness or as anxiety. As we interact with others we may try to infer their emotional states relying on subtle cues. We incrementally adjust our conclusions with every cue, moving closer to one emotion or the other, as make predictions and receive feedback during the interaction. If, by the end of the interaction, both emotions seem equally likely, then either the situation is uncertain or that the person is experiencing an emotion outside of those two. Another promising class of models used iterative optimization in task abstractions space to discover latent tasks and associate them with an internally generated task abstractions (Butz et al., 2019; Hummos, 2023; Sandbrink et al., 2024). While these models offered a solution for continual learning, they relied on rounds of iterative optimization in task abstractions space, and were based on a limited understanding of of gradient descent dynamics in this space. GBI tackles the computational efficiency by investigating a role for one-step gradients to infer task abstractions. Further, by using pre-defined human-labeled task abstractions we were able to assess the accuracy and uncertainty estimates of gradient-based inference in task abstraction space.

Our findings confirm that empirical findings from cognitive science do indeed extend to neural networks. During training, providing task abstractions leads to **faster learning** (improved data-efficiency), and **limits forgetting** by inducing task-specific modules in the underlying neural network. During test, with weights frozen, models trained with task abstractions show **adequate task inference** through GBI, and can be adapted rapidly to produce different responses to the same input with **better capacity to generalize**. We first show these results in an intuitive contextual switching task for which we know the underlying Bayesian generative model. We then consider image classification datasets starting with MNIST and scaling up to CIFAR-100 (image generation, section 3.2). Further, this image generation framework allows us to test the capacity for the **detection of out-of-distribution (OOD) samples** and **estimating uncertainty**. In a third experiment (language experiment, section 3.3), we show that GBI is a domain-general method, and demonstrate several of these results in language modeling, specifically, task inference, data efficiency, and generalization.

## 2 METHODS

**Overview.** We assume a dataset is equipped with a task abstraction. Conceptually, a task abstraction groups data samples into conceptual categories defined across dataset samples rather than individual data points. Mathematically, we model this by a graphical model where the data point $\mathbf{X}$ is generated from a task abstraction $\mathbf{Z}$. Now we use a neural network to model this graphical model with data $\mathbf{X}$, task abstraction $\mathbf{Z}$ and unknown parameter $\theta$, our neural network model has the form of a joint likelihood function

$$\mathcal{L}(\theta; \mathbf{X}, \mathbf{Z}) = p(\mathbf{X}, \mathbf{Z}|\theta). \tag{1}$$

At the training phase, given a data point $\mathbf{X}$ with the abstraction of interest $\hat{\mathbf{Z}}$ directly observed, we train the neural network by doing gradient descent on $-\log \mathcal{L}(\theta; \mathbf{X}, \hat{\mathbf{Z}})$ with respect to $\theta$.

At the test phase, we no longer assume access to task abstraction $\mathbf{Z}$ and require the model to identify them in individual data points efficiently. Moreover, we require for our model a mechanism to manipulate the abstract representations to adapt to new unseen situations. Specifically, we update our $\mathbf{Z}$ through gradient descent on $-\log \mathcal{L}(\theta; \mathbf{X}, \mathbf{Z})$.

The prediction of the model with task abstraction $\mathbf{Z}$ is

$$\hat{\mathbf{X}} = \arg \max_{\mathbf{X}} \mathcal{L}(\theta; \mathbf{X}, \mathbf{Z}). \tag{2}$$

**Connections to Expectation-Maximization algorithm.**   Notice that this method has deep connections to the classical Expectation-Maxmimization (EM) algorithm. In the training phase, since $\hat{\mathbf{Z}}$ is drawn from the distribution $p_{\mathbf{Z}|\mathbf{X}}$, the training attempts to find $\hat{\theta}$ such that

$$\hat{\theta} = \arg \max_{\theta} \mathbb{E}_{p_{\mathbf{Z}|\mathbf{X}}}[\log \mathcal{L}(\theta; \mathbf{X}, \mathbf{Z})]. \tag{3}$$

Equation 3 is a compact way to write the classical EM algorithm where the expectation steps are replaced by direct observation. Let the probability distribution of $\mathbf{Z}$ be $q$. We can write

$$\log p(\mathbf{X}|\theta) = \mathbb{E}_q[\log p(\mathbf{X}|\theta)] \tag{4}$$
$$= \mathbb{E}_q[\log p(\mathbf{X}, \mathbf{Z}|\theta) - \log p(\mathbf{Z}|\mathbf{X}, \theta)]. \tag{5}$$

By adding and subtracting an entropy of $q$, we have

$$= \mathbb{E}_q[\log p(\mathbf{X}, \mathbf{Z}|\theta)] + H(q) + KL(q||p_{\mathbf{Z}|\mathbf{X}}) \tag{6}$$

where KL divergence $KL(q||p)$ is defined as $-\mathbb{E}_q[\log \frac{p}{q}]$ and the entropy $H(q)$ is defined as $-\mathbb{E}_q[\log q]$. Now the expectation step corresponds to find $q$ such that $\mathbb{E}_q[\log p(\mathbf{X}, \mathbf{Z}|\theta)] + H(q)$ is maximized. Notice that this is equivalent to minimizing the KL divergence $KL(q||p_{\mathbf{Z}|\mathbf{X}})$ and we know it is minimized at 0 when $q = p_{\mathbf{Z}|\mathbf{X}}$. In particular, we have $\mathbb{E}_q[\log p(\mathbf{X}, \mathbf{Z}|\theta)] + H(q) = \log p(\mathbf{X}|\theta)$.

At the maximization step, we find $\theta$ such that $\mathbb{E}_q[\log p(\mathbf{X}, \mathbf{Z}|\theta)] + H(q)$ is maximized, but since $H(q)$ does not depend on $\theta$, it is equivalent to maximize $\mathbb{E}_q[\log p(\mathbf{X}, \mathbf{Z}|\theta)]$ and therefore it is exactly Equation 3.

At our testing phase, since we are not able to access $p_{\mathbf{Z}|\mathbf{X}}$ directly, we optimize $q$ through gradient descent. Our objective is exactly the expectation step with the regularization term $H(q)$ implemented as L2 regularization on $\mathbf{Z}$.

There are different methods to optimize $\mathbf{Z}$ and we discuss two methods we use in this paper, iterative optimization and one-step gradient update below to obtain gradient-based inference (GBI) estimates over latent variable values.

**Iterative optimization of Z.**   One method for doing gradient descent on $\mathbf{Z}$ is to iteratively optimize $\mathbf{z}$ using the gradient $\partial \mathcal{L}/\partial \mathbf{z}$. We implement iterative optimization using Adam optimizer with a learning rate of 0.01 and L2 regularization scaled by 0.01. This method usually takes many iterations (Marino et al., 2018). Another method is to learn an update rule $f_\phi(\cdot)$ that converts the gradient directly into an inferred $\mathbf{z} \leftarrow f_\phi(\partial \mathcal{L}/\partial \mathbf{z})$ (Marino et al., 2018). This method allows for rapid inference during run time but requires training of the update rule (like other meta-learning methods).

**One-step gradient update.**   We explore an alternative that requires only one pass through the model by updating the gradient only once. Without iteratively optimizing $\mathbf{Z}$ through gradient update, one-step gradient update usually heavily depends on the initialization: because if the initialization is far away from the optimal $\mathbf{Z}$, the local gradient information might not be informative of the global optimum. However, notice that in the expectation step, our goal is to find the probability distribution of $\mathbf{Z}$, $q$, such that $\mathbb{E}_q[\log p(\mathbf{X}, \mathbf{Z}|\theta)] + H(q)$ is maximized. We can consider an alternative objective function $f_\alpha(q) = \mathbb{E}_q[\log p(\mathbf{X}, \mathbf{Z}|\theta)] + \alpha H(q)$ for some $\alpha > 0$ and let $q_\alpha^*$ be the point that maximizes this alternative objective function. Since entropy is concave, by increasing $\alpha$ we make the function more concave and therefore the gradient is more informative of the actual optimum (Fig 1B). Furthermore, $q_\alpha^*$ will also become closer to the maximal entropy point $\bar{q} = \arg \max_q H(q)$. Therefore we can choose our initialization point as the maximal entropy point and output $\hat{q}$ using our one step gradient update rule:

$$\hat{q} = \text{softmax}(\bar{q} + \nabla f_\alpha(\bar{q})). \tag{7}$$

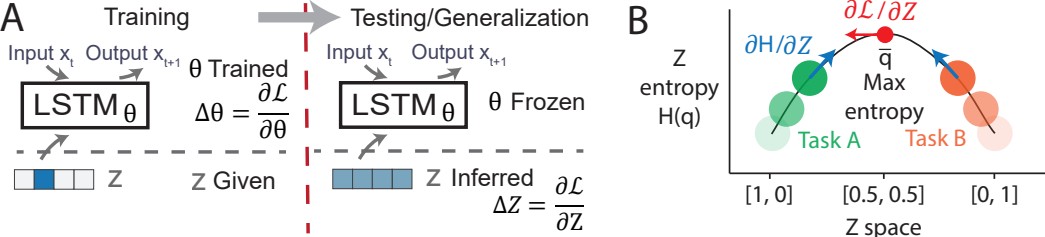

Figure 1: Schematics of the framework. A) Schematic of training phase and testing/generalization phase. During training, ground truth Z provided to the model and neural parameters $\theta$ are optimized. During testing, $\theta$ remains fixed, while Z is inferred using gradient descent. B) Schematic of how regularization of Z during training shapes the task abstraction space. Z values for Task A and B are pushed towards the maximum entropy point in Z, and are organized on the entropy curvature around the max entropy points. Gradient steps from the maximal entropy points then push the model towards the Z values of the appropriate task representation.

Here, we use a softmax function to project $q$ back to the space of distribution. We implement this update using the SGD optimizer with learning rate 1 and no momentum terms. We reset our initialization point to maximal entropy to obtain $\hat{q}$ for the next data point.

In the following section, we implement these methods in a toy task (for comparison to Bayesian inference), an image generation and classification model, and finally, a simple language model. Code available at `https://anonymous.4open.science/r/neuralbayes-F06F/`.

## 3 EXPERIMENTS

The experiments are organized as follows. In Section 3.1, we first investigate the properties of GBI on a simple synthetic dataset generated from a ground truth Bayesian model. We show that GBI displays better data-efficiency, generalizes better and forgets less, in addition to being able to pass neural gradients back to the Bayesian model to support Bayesian computations. Section 3.2 assesses the potential of GBI as a novel image classifier, the GBI ability to generate task samples, and to detect OOD samples. Finally, in Section 3.3, we demonstrate the abilities of GBI to recompose previously learned tasks, and thereby flexibly adapt to novel tasks using a language model.

### 3.1 EXPERIMENT 1, TOY TASK: TASK ABSTRACTIONS REDUCE FORGETTING AND IMPROVE GENERALIZATION BY INDUCING TASK MODULES

We begin with a simple dataset of one-dimensional observations unfolding in time, $\mathbf{x}_{0:t}$. The observations are generated from two alternating Gaussian distributions with distinct means ($\mathcal{N}(0.2, 0.1)$ or $\mathcal{N}(0.8, 0.1)$, Fig 6A, C). The ground truth generative causal model (Fig 6A) has the task node ($\mathbf{z}$) as a binary variable with values either $\mathbf{z}^1 = [0, 1]$ or $\mathbf{z}^2 = [1, 0]$. Context node switched between those two values with a probability $P_v = 0.005$, but we enforced a minimum block length of 20 and a maximum of 50 (further details in Appendix B). Knowing the ground truth generative model, we can analytically calculate the likelihood of each $z^i$ as $p(x_t|z^i)$ and the posteriors $p(z^i|x_{0:t})$(Fig 6C).

To estimate these Bayesian quantities from a neural network, we train a 100-unit LSTM (Hochreiter & Schmidhuber, 1997) to predict the next observation $x_{t+1}$ given: (i) the five previous observations ($\{x_t, ..., x_{t-h}\}$, h = 5) presented sequentially, and (ii) a task abstraction input which we set to the task z values, one-hot encoded with either [0, 1] or [1, 0] after passing through a softmax function. We name this model GBI-LSTM and train it on the task (training details in appendix B), and in the following sections, we compare its learning dynamics to an LSTM with no such task input.

**GBI-LSTM is data efficient.** GBI-LSTM learns the task faster (i.e. better data efficiency) (Fig 2C,D). Example full runs in appendix B (Fig S9), (LSTM, Fig S8). On one hand, this seems obvious as the GBI-LSTM gets additional contextual information, but on the other hand, the task is simple enough that it is not clear that a 100-unit LSTM needs the additional information. To explain, we

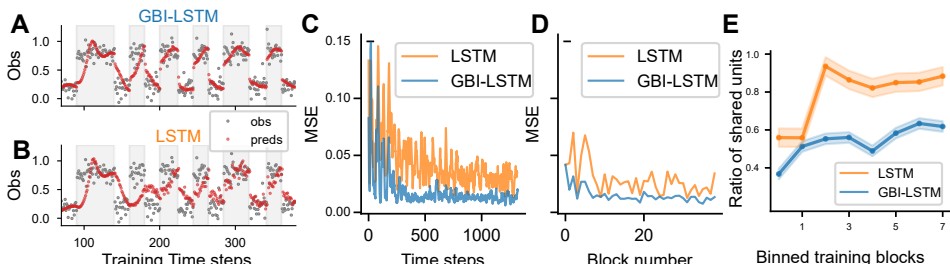

Figure 2: Training on a toy sequence prediction task with and without task abstractions input. One-dimensional data is generated from a Gaussian distribution that flips its mean every 20-50 timesteps (between $\mu_1 = 0.2$ or $\mu_2 = 0.8$, with fixed standard deviation $\sigma$ at 0.1). An example training data for A) an LSTM trained with task input (GBI-LSTM) and B) without task input (LSTM). C) We compare the training loss of GBI-LSTM and LSTM. Note the variability in these loss estimates despite the simulation run over 10 random seeds is due to the stochastic block transitions still coinciding frequently. D) We show the mean loss per block but exclude the initial 20 predictions in each block to exclude poor performance from either model limited to transitioning between tasks. E) We quantify the ratio of shared neurons active in the 0.2 blocks and the 0.8 blocks using 'task variance' to identify engaged neurons. We binned training blocks into groups of 5 to aggregate the data and increase accuracy.

reasoned that the available task abstraction deeply influences what solutions the network learns and allows for modules to emerge for each task. We used 'task variance' to identify neurons active in each task, a measure used in neuroscience literature (Yang et al., 2019). We identify active neurons by selecting units with the highest variance across samples of inputs and outputs for a task. We see that neurons for each task overlap in the LSTM but separate into modules in the GBI-LSTM, throughout training (Fig 2E). At *inference time*, with weights frozen, the baseline LSTM handles task transitions efficiently (Fig S8). The GBI-LSTM has the weights frozen but we now iteratively optimize the task abstraction layer z using vanilla SGD optimizer with a learning rate of 0.5 and we see that it also transitions between tasks by updating its task abstraction input z (Fig 3A-C, Fig S9).

**GBI-LSTM generalizes better and forgets less.** GBI-LSTM generalizes better to values outside of the training range (Fig 3D). By using iterative optimization, a gradient step in **Z** every time step, the GBI-LSTM can interpolate and better predict data points drawn from other distributions far from the two training distributions (Fig 3E). Moreover, we observe that the baseline LSTM already shows signs of catastrophic forgetting in this very simple setting. Testing MSE is worse around one of the two training means depending on which mean generated the last block of data during training. In figure 3D, we show the responses from runs where the last training block had the 0.2 Gaussian mean active. In contrast, as quantified in Table 1 the GBI-LSTM shows no signs such forgetting (Fig 3E).

Table 1: Quantifying the generalization and forgetting on the toy task. Both models were trained on data generating means 0.2 and 0.8. We aggregate model prediction errors (mean MSE values $\pm$ SEM) tested on data generated from means -0.2 through 1.2. We identified the data generating mean during the second last block of the training sequence to assess forgetting. 20 runs with different seeds.

| Data range | LSTM MSE | GBI-LSTM MSE |
|---|---|---|
| Second last training block (0.2 or 0.8) | $0.30 \pm 0.03$ | $\mathbf{0.24} \pm 0.02$ |
| Inside training range (0.3-0.7) | $0.27 \pm 0.02$ | $\mathbf{0.25} \pm 0.01$ |
| Outside training range ($<0.2$ & $>0.8$) | $0.35 \pm 0.06$ | $\mathbf{0.26} \pm 0.03$ |

**Bayesian properties of GBI-LSTM.** We can use our GBI-LSTM for inference, i.e. approximate Bayesian quantities such as the posterior and the likelihood function of z. We distinguish between two ways we can use the GBI-LSTM, first by taking one-step gradient updates from maximal entropy points ('default state' mode) and second is the iterative optimization of the task abstraction z at a lower learning rate ('task-engaged' mode) which we used above to perform the task and generalize to novel data. In the 'default state' mode, we set the task abstraction layer to its state of maximum

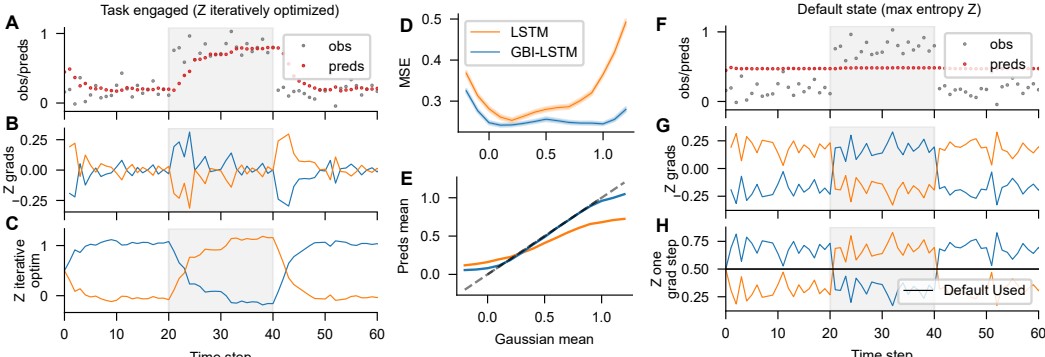

Figure 3: Neural network trained with task abstractions can dynamically tune itself to incoming data and generalizes to novel data points. A) Responses of the GBI-LSTM to sample data and B) the gradients of the prediction loss w.r.t. to the two Z units. C) Iteratively optimized Z values accumulating evidence dynamically. D) As a generalization test, while the network was trained on data from Gaussians with means 0.2 and 0.8, we test its responses to Gaussians between -0.2 to 1.2 (in 0.1 increments). We record the mean MSE values across 20 runs with different random seeds and compare to the baseline LSTM trained without task abstraction input. We only show runs where the last block of training was a 0.2 block, to highlight models behavior on that mean vs the other (0.8). E) The mean of the Gaussian data vs. the mean network predictions in each generalization block. These results are further quantified in Table 1. F-H) We show the other mode of using the same GBI-LSTM for gradient-based inference. We fix the task abstraction input Z to its state of maximum entropy (here [0.5, 0.5]) and take gradients from that point. While the network responds in the middle (F), the gradients w.r.t to Z (G) or similarly, the one-step gradient descent Z values behave much like the likelihood function, c.f. values computed using a Bayesian graphical model 6C.

entropy (here, the mean of the two values observed in training). For each time step, we take one step in z space to lower the prediction error, and then reset to max entropy for the next point. These one step z values (Fig 3H) behave much like the likelihoods of z computed through the ground truth causal model (See Fig S6 for a plot of the likelihoods and one gradient step z overlaid). Moreover, we see that these one step z can support Bayesian computations when used in lieu of the likelihood term in the Bayesian equation (previously calculated using the Gaussian probability distribution equation). i.e., we are able to pass one step z values to a Bayesian graphical model and Bayesian computations proceed normally (Fig S7).

### 3.2 EXPERIMENT 2, IMAGE GENERATION: ONE-STEP GRADIENTS FOR ACCURATE TASK INFERENCE AND OOD DETECTION

We next assess how accurate GBI is in inferring task abstractions in an image generation setup with more tasks. In addition, we show that GBI has more beneficial properties, such as better OOD detection, and uncertainty estimates that require no calibration.

We train a convolutional autoencoder to reconstruct images. The decoder gets an additional input of a one-hot encoding of the image class in addition to the latent embedding from the encoder (Fig 4 Schematic). We train this structure for 4 epochs using an MSE reconstruction loss (full details in Appendix C). Confirming the results observed in the toy task, the network receiving additional image class trains faster compared with baseline autoencoder, (Fig S12).

**One-step gradients infer task abstractions with sufficient accuracy.** At test time, we set the task abstraction (i.e., image class) input z to maximum entropy point and take one gradient step of the reconstruction loss with respect to the class inputs z, and directly interpret the resulting vector to infer the class label. We make two kinds of comparison, first to methods that used gradient-based inference, we refer to them as entropy-based (Wortsman et al., 2020) and norm-based inference(Roy et al., 2024) and we see that GBI outperforms previous gradient-based methods (Table 2, implementation details in appendix C). Second, we compare GBI with four canonical methods to infer class based on the same architecture, as follows: **(i)** a convolutional classifier of the same architecture as the encoder, as

a measure of what one might gain from a dedicated inference model that maps from inputs to task abstraction (in this case, image class). **(ii)** iteratively optimizing the latent following the traditional variational inference paradigm (VI), which is the same as our model, but increases the computational budget by taking many smaller steps of gradient descent in task abstraction space. **(iii)** evaluating the model's MSE loss at each image one-hot encoding and selecting the one with lowest MSE loss, as a general representation of Bayesian methods that require evaluating a likelihood function over all hypotheses of interest. **(iv)** evaluating the model's reconstruction MSE again, but in a purely generative model trained from task abstraction input to image reconstruction with no encoder, as an ablation to the separation of task input and task abstraction in our model. The goal: show that GBI suffers only a small drop in accuracy while offering a small computational budget and no added parameters. We see that GBI outperforms other gradient-based methods EBI and NBI, and show only a small drop in accuracy compared to canonical classifier (Table 2). We can also use our same model and increase the computational budget with iterative optimization or serial likelihood assessments when higher accuracy is required. More importantly, GBI, being a generative model, offers a set of unique advantages as we discuss next.

**High OOD detection with GBI.** GBI retains many of the desirable properties of a generative model and thus can evaluate the probability of the data under the model. Therefore, we reasoned that GBI may estimate uncertainty accurately (Fig S10) giving it an advantage in detecting out-of-distribution (OOD) over traditional models. To test this, We compared a traditional convolutional classifier with GBI, both trained on MNIST digits and using fashionMNIST clothing items as the OOD dataset (Xiao et al., 2017). Using one gradient step task abstraction layer logits, GBI is superior at detecting OOD samples compared with the convolutional classifier (figure4). In comparing our OOD detection results to current state-of-the-art methods we noted that the standard setup in studying OOD detection does not involve normalizing the image brightness mean and variance, making it trivial to discriminate the in and out-of-distribution

Table 3: One gradient step values in GBI trained MNIST show better OOD detection on fMNIST dataset compared to classifier logit and state-of-the-art likelihood regret method. AUCROC values averaged over 10 seeds with standard deviations. We examine the case when pixel intensity statistics were normalized so methods are not trivially detecting changes in brightness. Ensemble networks and Bayesian neural networks though perform better that likelihood regret in this setting. Unnormalized results in Tab S6

| Method | AUCROC |
|---|---|
| GBI | **0.89** $\pm$ 0.03 |
| Classifier Softmax Maxes | 0.73 $\pm$ 0.08 |
| Likelihood Regret | 0.80 $\pm$ 0.06 |
| Ensemble networks | 0.809 $\pm$ 0.011 |
| Bayesian Neural nets | 0.859 $\pm$ 0.040 |

datasets. We compared our OOD results to the Likelihood Regret (LR) method for OOD detection in a generative model Xiao et al. (2020) (see supplementary material for implementation details). On unnormalized data samples, where detection methods can take advantage of differences in pixel intensities between the in and out-of-distribution samples, both GBI and LR perform high (Fig A14, Table A6). However, when data samples are normalized GBI shows a clear advantage over LR (table 3).

Lastly, we provide an illustration of how the GBI framework enables adaptive behavior, where in response to the same input (image), changing the task abstraction (the image label) can lead to a different input-to-output transformation, allowing for an additional layer of flexibility in task selection (Fig S11).

**CIFAR-100.** We apply GBI to the CIFAR-100 dataset, which features color images of vehicles, animals with diverse backgrounds. The GBI accuracy 18% well above that obtained from a pure generative model (Table 4) (i.e. a model to decode image class to image directly). This suggests a positive effect from the separation of visual input and task abstraction streams. We see only a small drop between evaluating the likelihood of each image class for each image (requires one for each of the 100 image classes, accuracy 21%) and GBI which requires only 1 run. However, GBI accuracy is rather low, suggesting that backpropagation found solutions that relies on the visual input features more than the image class input. This observation coupled with the overall low accuracy from the pure generative model suggests that image class input does not significantly reduce the variance seen in pixel space, leading the decoder to favor the visual features from the encoder. We anticipate that scaling to larger datasets will benefit from richer task abstractions. This highlights a role for algorithms that discover richer abstractions directly from data (Butz et al., 2019; Hummos, 2023).

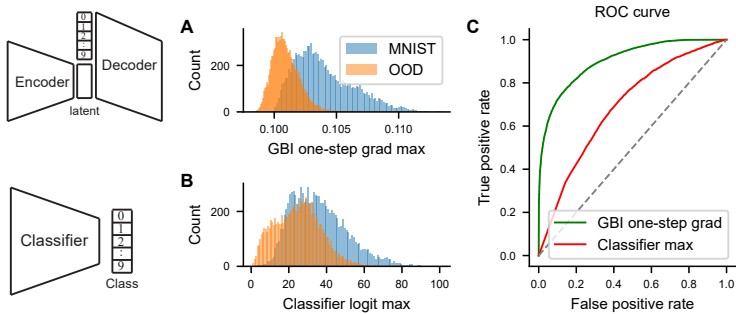

Figure 4: Robust out-of-distribution (OOD) samples detection in a GBI autoencoder compared to a classifier. Both models were trained on MNIST digits, and tested on OOD dataset fashionMNIST. We take the A) max value of the task abstraction layer in GBI and for the B) logits in the classifier and consider how are they distributed for in-distribution samples and OOD samples. C) ROC curves show that GBI max values are more discriminative.

## 3.3 EXPERIMENT 3, LANGUAGE: FASTER TRAINING AND ACCURATE TASK INFERENCE AND GENERALIZATION

Next we consider language modeling for a self-supervised sequence modeling task. During training, we examine the effects of task abstractions on data-efficiency, and during testing, we evaluate ability to infer task abstractions and ability to generalize to unseen language datasets.

**Method** We train an LSTM (with 2 layers, 200 units each) on word-level language prediction and compare an LSTM with our GBI-LSTM. To demonstrate the effectiveness of task abstractions in this setting, we represent each task abstraction as a one-hot encoded identifier of each training dataset. We use datasets included in the BabyLM challenge (Warstadt et al., 2023), which provides 10 datasets inspired by what language data children might be exposed to during development, including wikipedia dataset, movie subtitles, children stories, and adult conversations, amongst others (listed in Appendix E). We train the two models on 3 of these datasets in a setting analogous to the Bayesian model dataset described earlier with data in each batch randomly drawn from one of the three datasets (we vary the choice of the 3 training datasets across seeds). The GBI-LSTM receives additional input with a one-hot encoded dataset identifier during training concatenated to the input token at each time step.

Table 4: GBI accuracy on CIFAR100 using a multiplicative task abstraction. The task abstraction is projected through a frozen embedding layer ( Bernoulli distribution) then multiplies the visual information going to decoder.

| Method | Test Accuracy (%) | Model runs |
|---|---|---|
| Pure generative | $9.57 \pm 0.19$ | 100 |
| GBI | $18.52 \pm 0.38$ | 1 |
| Iterative optimization | $18.53 \pm 0.38$ | 400 |
| Likelihood | $21.30 \pm 0.36$ | 100 |

Table 2: GBI outperforms other gradient-based inference methods, and compares well to other canonical methods in ML (small drop in accuracy, but only one run through the neural network component). For iterative optimization, we further examine the relationship between optimization steps and accuracy in Fig S13.

| Method | Accuracy (%) | Runs |
|---|---|---|
| **Canonical methods** | | |
| Classifier | $91.44 \pm 0.51$ | 1 |
| Pure generative | $81.91 \pm 2.3$ | 10 |
| **Ours** | | |
| GBI | $85.46 \pm 3.60$ | 1 |
| Iterative Optimization | $88.51 \pm 3.42$ | 50 |
| Likelihood (Recon MSE) | $91.94 \pm 3.38$ | 10 |
| **Other gradient-based methods** | | |
| EBI | $27.37 \pm 2.22$ | 1 |
| NBI | $78.78 \pm 1.21$ | 10 |

During testing, we use one-step gradients to infer the dataset from the text data provided as we assume no access to the dataset identifier at test time. We then use iterative optimization of the task abstraction input z to adapt the model to novel datasets by taking gradients steps on one batch of data and evaluating on the test set.

**Results** First, we focus on the effect of task abstractions on data-efficiency. Compared with the LSTM, GBI-LSTM displays lower training and validation losses (Fig 5A, B). While the improvement is modest, we emphasize that we only had to add <1.6 bits of information to obtain it.

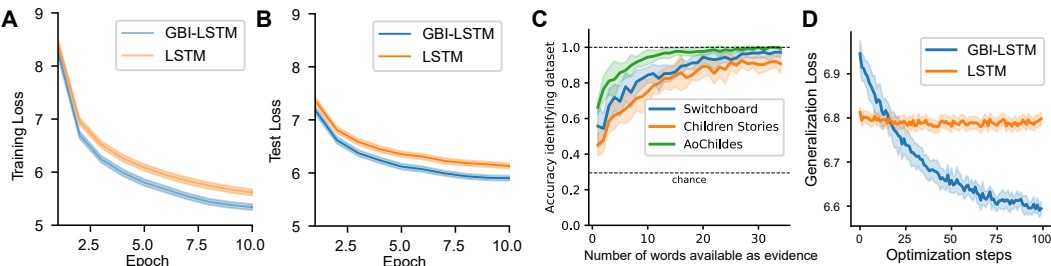

Figure 5: Better data-efficiency in an LSTM trained with task abstraction input (dataset ID one-hot) (GBI-LSTM) compared to training without such input (LSTM). Cross entropy loss for GBI-LSTM decreases faster for GBI-LSTM on the A) training set and B) testing set. Note that we train with a word-level tokenizer with a larger output dimension which has a highe cross entropy loss upper bound ( 12). Each simulation run had three language datasets picked randomly from 10 datasets. 10 data set picks, each run for 3 random seeds, shaded area SEM. C) One-pass gradient-based inference can infer dataset ID accurately at test time needing only a few words. D) Iterative optimization can be used to adapt a GBI-LSTM to a novel dataset. We compare it to optimizing the inputs of an LSTM that was trained with no task provided to rule out any generic optimization effects. Due to the variability of loss values across datasets, for each choice of 3 training datasets (12 sets), we take the mean and SEM over 4 seeds and then aggregate the results from all 48 model runs.

Second, we focus on task inference. We show that GBI can quickly and reliably infer the dataset at hand, and in fact, it takes a few words from the first batch to accurately infer the dataset (Fig 5C). Note that similar to likelihood functions, gradient inference can handle variable amounts of evidence, and the estimation accuracy scales with the amount of evidence available.

Third, we focus on generalization. Our results show that task abstraction layers improves the model's ability to generalize to novel tasks (Fig 5D), similar to the case in the synthetic Bayesian dataset (Fig 3D,E). We use a 4th dataset not seen during training, and we compare the GBI-LSTM and LSTM losses on this novel dataset. To allow gradient descent to compose the closest solution to the novel datasets, we optimize the task abstraction input nodes of the GBI-LSTM to lower its prediction loss and we see a decrease in loss as it adjusts to this new unseen dataset (Fig 5D) (See Appendix E for example run). Notably, the GBI-LSTM starts out at a higher loss, on average, highlighting that task-aware models require task inference.

## 4 CONCLUSIONS AND FUTURE DIRECTIONS

Overall we show that neural networks with access to task abstractions learn faster, forget less, adapt to changes in data distribution, and generalize to new situations. At the same time, we also provide tools for the networks to function independently infer their own task abstractions during testing. These task abstractions can be provided by humans, a trained teacher model, or inferred from data with no supervision (e.g. (Hummos, 2023)). Task-dependent networks require task inference and inferring an incorrect task impairs function. In animal brains, some computations that are frequently needed or are critical for survival might be better mapped to the default state of the brain. The brain shows distinct dynamics when not engaged in a specific task, the default mode, and we show previously unexplored connections between feedback signals in this default state and distributional inference signals that can support many probabilistic quantities such as uncertainty and surprise. We see future work addressing the limitation of relying on human-provided task abstractions as opposed to unsupervised discovery of richer multidimensional task representations along dimensions important for adaptation and generalization. Finally, while our method is theoretically motivated, we see future work extending these findings to larger datasets and more real-world applications.

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

## A RELATED WORK

### A.1 ADAPTING LARGE MODELS

Several methods have been developed for adapting large models to specific tasks without expensive full fine-tuning. Prompt and prefix tuning involve the use of learnable prompts (Lester et al., 2021) or a continuous prefix (Li & Liang, 2021) that guide the model's outputs. Lower-rank adaptation (Hu et al., 2021) modifies model parameters with a low-rank tensor, learned during adapting the model, while Adapters introduce small, task-specific modules within the network that are trained while the rest of the model remains fixed (Houlsby et al., 2019). Different from these methods, our approach to contextualizing a generative model proposes providing context signals or inferring them during the pretraining of the model. This allows the model to organize knowledge along these dimensions giving it an inductive bias anticipating variations along the same dimensions.

### A.2 ITERATIVE VARIATIONAL INFERENCE

Variational inference, a method that recasts inference as an optimization problem (Jordan et al., 1998; Neal & Hinton, 1998), has been instrumental to the development of deep variational models. However, recent work has primarily used the variational formulation for training, relying on fast amortized directed encoders to infer an approximate posterior distribution (Kingma & Welling, 2013; Rezende et al., 2014), thereby avoiding the potentially lengthy optimization loops of taking gradients through the decoder to refine the posterior distribution. To mitigate this, amortized variational inference employs an additional neural network that takes either the gradients from the decoder, or the reconstruction errors and updates the approximate posterior distribution with only a few iterations

needed (Marino et al., 2018). Our work builds upon these insights, demonstrating that the initial gradient to the latent carries sufficient information to approximate likelihood functions. This reduces the need for multiple model runs or learning an additional model, but keeps iterative refinement as an option for complex samples.

### A.3 Quantifying Uncertainty

Uncertainty quantification is a critical aspect of both Bayesian and neural network paradigms, with a rich and extensive history of literature, for review see (Abdar et al., 2021). Many approaches derive from the maximum likelihood interpretation of neural network training (MacKay, 1992), known as Bayesian Neural Networks (BNNs). Such models model uncertainty by placing prior distributions over the model's weights, with Variational Inference and dropout-based methods (like MCdropout and MCdropConnect) serving as practical approximations (Blundell et al., 2015; Graves, 2011; Gal & Ghahramani, 2016; Srivastava et al., 2014). Ensemble methods, such as Deep Ensembles, improve predictive performance and provide calibrated uncertainty measures by aggregating predictions of multiple independently trained neural networks (Lakshminarayanan et al., 2017). Lastly, temperature scaling is a post-processing method used to calibrate the model's softmax outputs, aligning the model's confidence with its prediction accuracy (Guo et al., 2017).

### A.4 Out-of-Distribution (OOD) Detection

The role of out-of-distribution (OOD) detection is essential for the secure implementation of machine learning systems. This is due to the tendency of models to deliver erroneously confident predictions when faced with data that diverges from the distribution on which they were initially trained. Current OOD detection methods fall primarily into two classes: discriminator-based either the logit or the feature space (Hendrycks & Gimpel, 2018) or generation-based approaches which employ either the disparity in data reconstruction or the estimation of density in latent space (Nalisnick et al., 2019), with the appealing intuition that generative models capture the data distribution and can detect out of distribution samples. While one might suspect that our method works because it adds a generative objective, recent work showed that several families of generative models might assign a higher probability to out-of-distribution data than in-distribution ((Nalisnick et al., 2019; Fetaya et al., 2020)). Using their rich visual latent spaces, they can generalize easily to other datasets. Different that these models, our model separates computation of visual features into one stream and the task abstractions into another, instead of assessing OOD probability from the visual features, we selectively assess in the task abstraction space.

## B    Bayesian Toy Dataset

### B.1    Bayesian graphical model task

#### B.1.1    Bayesian generative model

The task involves a sequence of observations $x_t$ with a new observation at each time step $t$. The observations are generated from the following Bayesian model with context nodes z, a categorical variable, here 2-way (binary). Variable z is represented as one-hot encoded vector that selects the mean between two different configurations, while variance remains constant, as follows:

$$\mathbf{z}_0 = \text{random choice: } [0,1] \text{ or } [1,0] \tag{8}$$

with a transition probability matrix:

$$P(\mathbf{z}_t|\mathbf{z}_{t-1}) = \begin{bmatrix} 1 - p_v & p_v \\ p_v & 1 - p_v \end{bmatrix} \tag{9}$$

Where $P_v$ (=0.05) is the volatility of the context, or the probability of context switching every time step. We enforce a minimum of 20 trials per block and a maximum of 50.

$$P(x_t|\mathbf{z}_t = [0,1]) = \mathcal{N}(0.8, \sigma)$$
$$P(x_t|\mathbf{z}_t = [1,0]) = \mathcal{N}(0.2, \sigma) \tag{10}$$

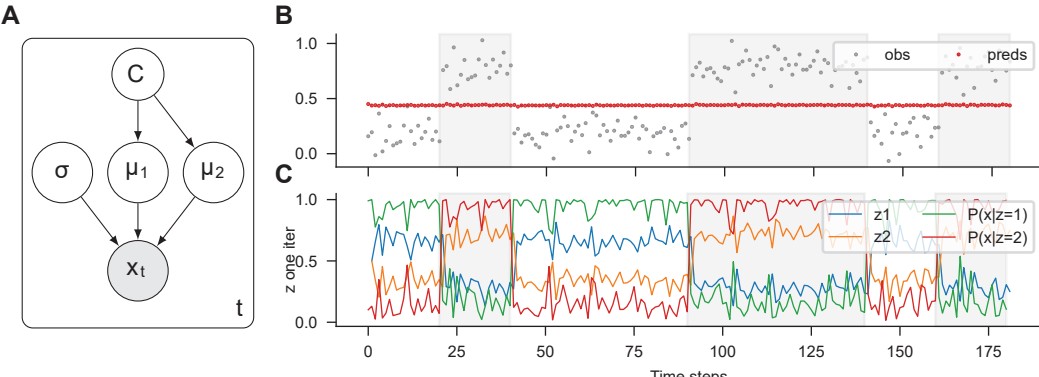

Figure 6: Comparing a Bayesian graphical model to GBI estimates of Bayesian likelihood of task abstractions. A) Bayesian graphical model generating the data. Task node (z) flips periodically and sets the mean of the Gaussian generating the data between $\mu_1 = 0.2$ or $\mu_2 = 0.8$, with fixed standard deviation $\sigma$ at 0.1. B) Sample generated data observations. The likelihood of each context node value given data is calculated using the Gaussian probability density equation, more details in Appendix B. B) GBI-LSTM was in default mode with z at max entropy as responses and one step gradients recorded. C) Overlaid likelihoods computed using the ground truth Bayesian model and neural gradients from GBI-LSTM for visual comparison.

Then using Bayes rule we express the posterior over context units at newest time step $\mathbf{z}_{t+1}$ given the history of the observations as follows:

$$P(z_{t+1}|x_{0:t+1}) = \frac{P(x_{t+1}|z_{t+1})P(z_{t+1}|z_{0:t}, x_{0:t})}{P(x_{t+1}|z_{0:t}, x_{0:t})} \tag{11}$$

and the likelihood function as the Gaussian probability distribution density:

$$P(x_t \mid z_t) = \frac{1}{\sqrt{2\pi\sigma^2}} \exp\left(-\frac{(x_t - \mu_1)^2}{2\sigma^2}\right) \tag{12}$$

### B.1.2 NEURAL MODEL ARCHITECTURE AND TRAINING

We employed the LSTM implementation provided by Pytorch, utilizing an input layer of size [1, 100] for mapping inputs to the LSTM, which consists of 100 hidden units. In the case of the GBI-LSTM the model received additional inputs from two additional units, the task abstraction units, after passing through a softmax activation function. An output layer of size [100, 2] was used to map to the output's mean and variance estimates. The model was trained to maximize the log-likelihood of the observed data by predicting a mean and a variance estimate, allowing the model to express a distribution over the next observation.. The final form of the objective derived from the Gaussian PDF has the MSE divided by the output variance. The LSTM was trained using the Adam optimizer with a learning rate of $10^{-3}$. To optimize the context representations, we employed Adam with a learning rate of $10^{-2}$ and a decay rate of $10^{-3}$.

## C  MNIST

### C.1  FURTHER RESULTS ON MNIST AND CONFIDENCE

We compared the confidence values from the GBI autoencoder trained on MNIST digits to a convolutional classifier. The classifier and the encoder shared the same convolutional network, and the decoder used its transpose convolutions. Being interested in the errors the models make, we limited the capacity of the model and used a small number of convolutional filters, specifically we used these layers, (table 5):

The classifier used this backbone followed by a non-linear layer mapping to 10 image classes, and then a softmax. While the GBI autoencoder produced the 8 dimensional embedding from the encoder,

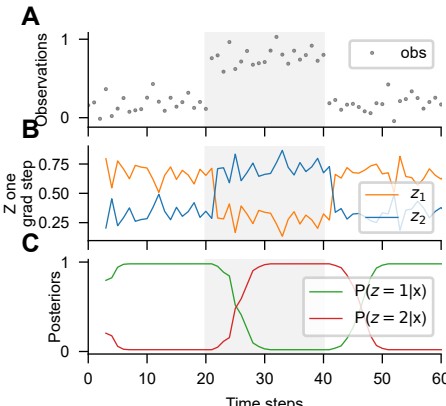

Figure 7: Grafting the neural gradients into Bayesian computations. With the task abstraction input z set to max entropy, we take one step along the gradients w.r.t to z. We use these one-step gradient updates in lieu of the likelihood function (eqn 12), thereby avoiding having to discover the $\mu$ and $\sigma$ nodes of the underlying Bayesian causal model, and offloading the structure learning to the neural network, but maintaining the ability to proceed with Bayesian computations.

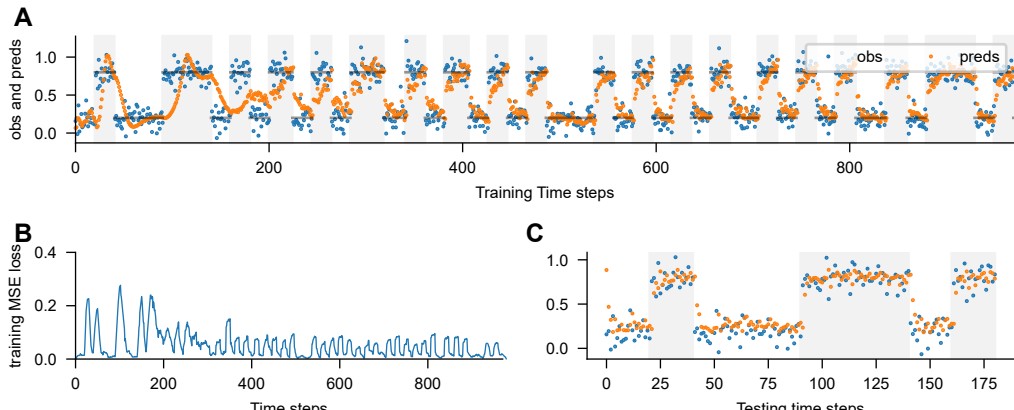

Figure 8: Training an LSTM on the Bayesian toy dataset, but no context from Bayesian nodes. An RNN trained on next observation prediction on this simple task learns it trivially well, and can adapt by detecting Gaussian distribution changes at context boundaries. Makes only one mistake typically before switching its predictions.

Table 5: Architecture of the convolutional classifier and encoder networks. Decoder had the same structure but transposed.

| Layer | Input Channels | Output Channels | Stride | Activation |
|---|---|---|---|---|
| Conv2d | image_channels | 2 | 2 | ReLU |
| Conv2d | 2 | 4 | 2 | ReLU |
| Conv2d | 4 | 8 | - | None |
| Flatten | 8 | - | - | - |

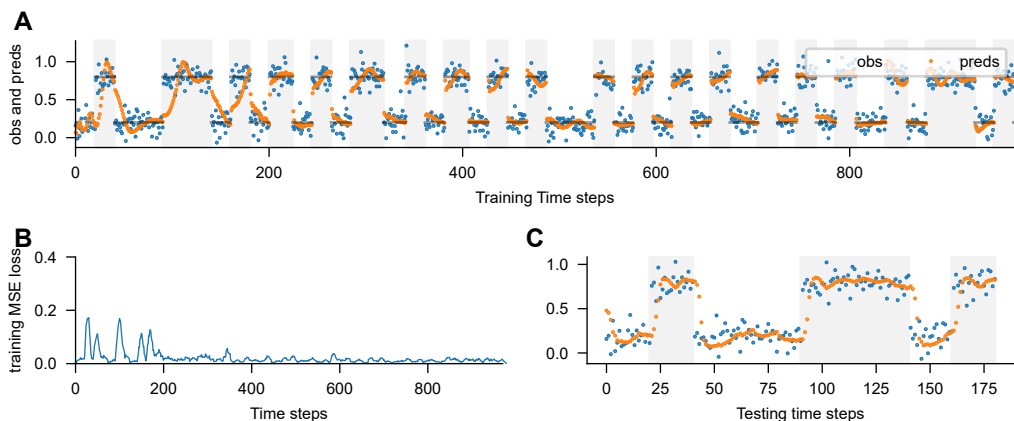

Figure 9: Training a GBI-LSTM on the Bayesian toy dataset with context from Bayesian model.

and concatenated the image label one-hot vector and passed those to the decoder. Of importance, we limited the encoder embedding output to only 8 dimensional vector, otherwise the decoder would ignore the labels and use the rich, but low level visual information from the encoder. We generated a few samples from the network by providing an original image from the test set, but changing the image label from 0 through 9, and examined the conditionally generated images, to ensure that the decoder was using the image label (Fig 11).

We examined the distribution of confidence values that GBI autoencoder produced, and found an informative distribution that can support post-hoc decision boundaries to get specific levels of accuracy, including 100% accuracy in the highest confidence bins. While a classifier trained with a softmax produces a distorted distribution that does not appear informative when binned (Fig S10). However, using the test set, one can post-hoc calibrate the softmax temperature of the classifier to obtain informative uncertainty estimates (Guo et al., 2017). Accordingly, our observation is that GBI does not require any post-hoc re-calibration to get an informative confidence distribution, but such as distribution is not unique.

**Other gradient-based methods details.** First methods is norm-based inference(Roy et al., 2024). These authors use the gradients norm to infer the task. They compute the gradient norm for each possible task label and select the one with the lowest norm. Second, we compared our method to entropy-based gradient inference (EBI), following the work of Wortsman et al. (2020). They run a forward pass based on the superposition of all task labels each weighted by an $\alpha$ value. Subsequently, they compute the output layer activation entropy, and take a step in $\alpha$ such as to minimize that entropy. To perform a one-iteration inference, they select the supermask (or task) with the largest gradient to minimize the entropy. Given that this method is limited to cross-entropy loss we reframe image generation as minimizing a cross-entropy pixel-wise classification loss.

## D    OOD DETECTION

We provide the GBI gradient and classifier logit maximum values and ROC in the case where the OOD dataset was not normalized. i.e., the OOD images had a different image brightness and variance (Fig S14), but we argue that an ideal image system should ignore these broad changes in illumination rather than rely on them to detect OOD samples.

**Likelihood regret implementation** We adapt the original code (https://github.com/XavierXiao/Likelihood-Regret.git) to the MNIST/FMNIST OOD detection task, using the suggested hyperparameters they used for the FMNIST-MNIST experiment since we are only testing for the opposite task. We ran 10 experiment seeds to calculate our results. We ran the tests with the two datasets pixel intensities normalized.

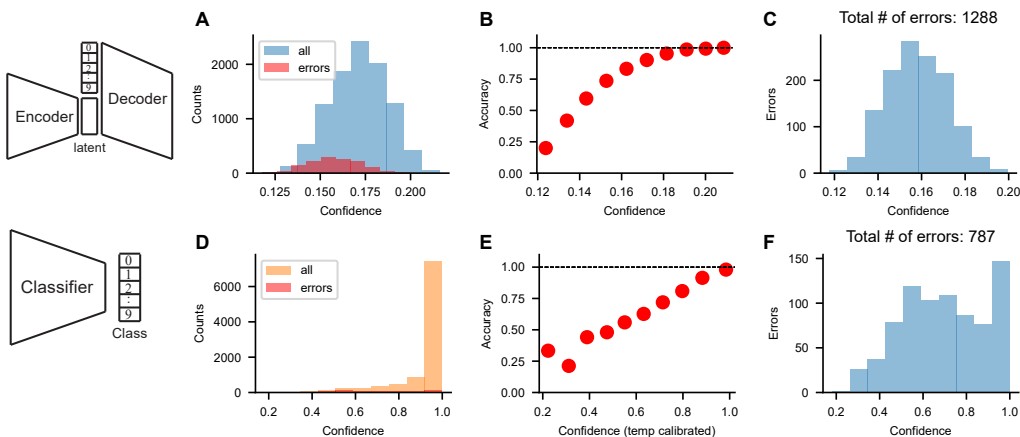

Figure 10: Confidence distributions and relationship with accuracy comparing a GBI network trained with task abstractions and a convolutional classifier.

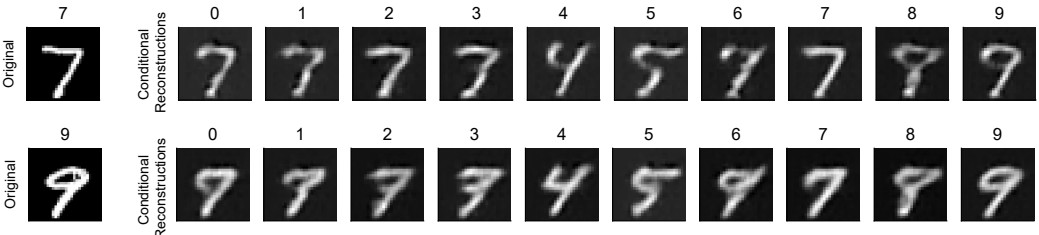

Figure 11: Conditional generation from the GBI autoencoder network. We feed an original image to the encoder and ask the decoder to reconstruct the image, as we vary the digit label input 0 to 9.

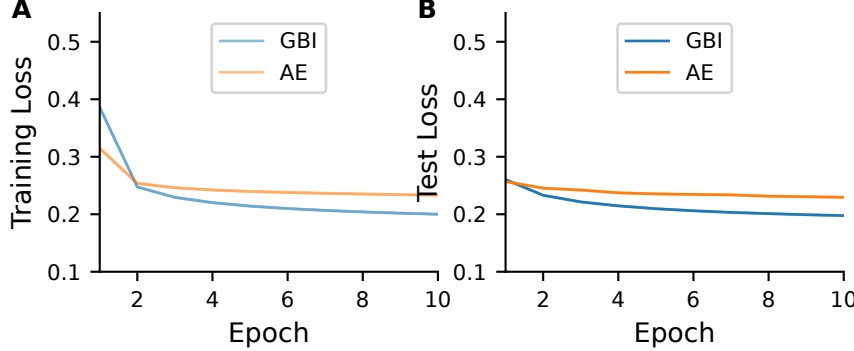

Figure 12: Comparing task-aware (GBI) and context-free autoencoder (AE) training and validation error on MNIST.

|  | AUCROC (Unnormalized) |
| --- | --- |
| **Method** | **Value** |
| GBI Maxes | $0.90 \pm 0.05$ |
| Classifier Softmax Maxes | $0.68 \pm 0.03$ |
| Likelihood Regret | $1.00 \pm 0.00$ |

Table 6: Same comparison as in table 3, but with the in-distribution and OOD datasets having different brightness statistics.

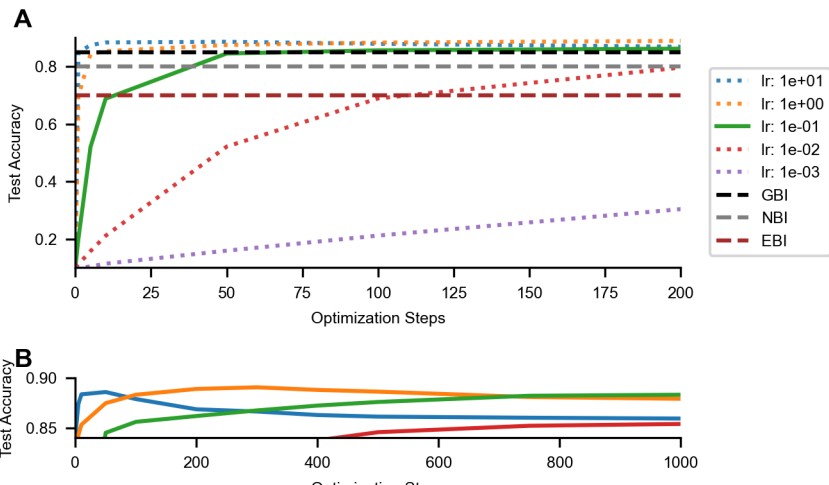

Figure 13: Variational inference iterative optimization steps vs accuracy. The models were randomly initialized in a classical variational inference fashion, and the latent representation was trained using vanilla SGD, which proved to be more stable than other optimization algorithms such as Adam and allowed control over learning rates. A) We highlight the accuracy line with the best end accuracy, other learning rates in dashed lines. Horizontal lines mark the three gradient-based methods compared in Table 2. B) A closer look at how higher learning rates eventually lead to lower accuracy, and we picked the best learning rate based on end accuracy being the higest.

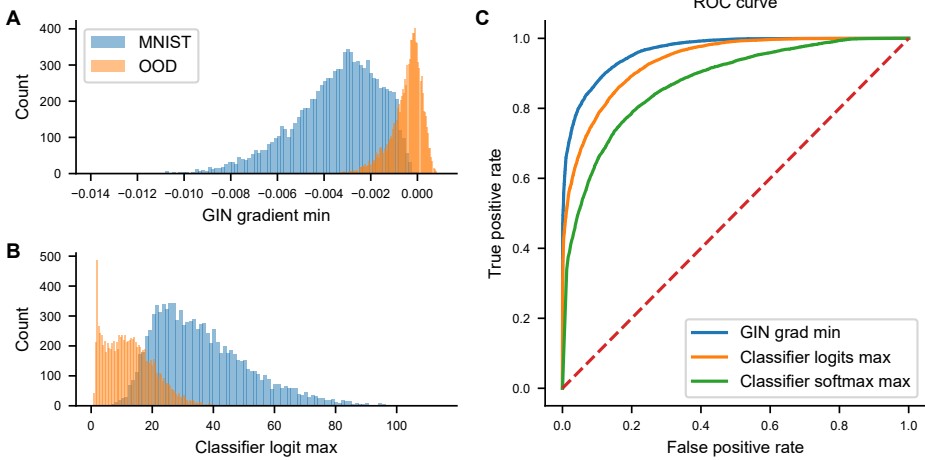

Figure 14: The figure compares the minimum gradient values of the (A) gradient based inference network (GBI) and the maximum classifier logits when fed an in-distribution image (MNIST) versus an out-of-distribution (OOD) image (fashionMNIST). B) The OOD images have a darker average intensity compared to MNIST, resulting in lower responses in the classifier logits. Exploiting this discrepancy, a discriminating threshold based on classifier logits yields a reasonable ROC curve (C) for distinguishing between the two distributions.

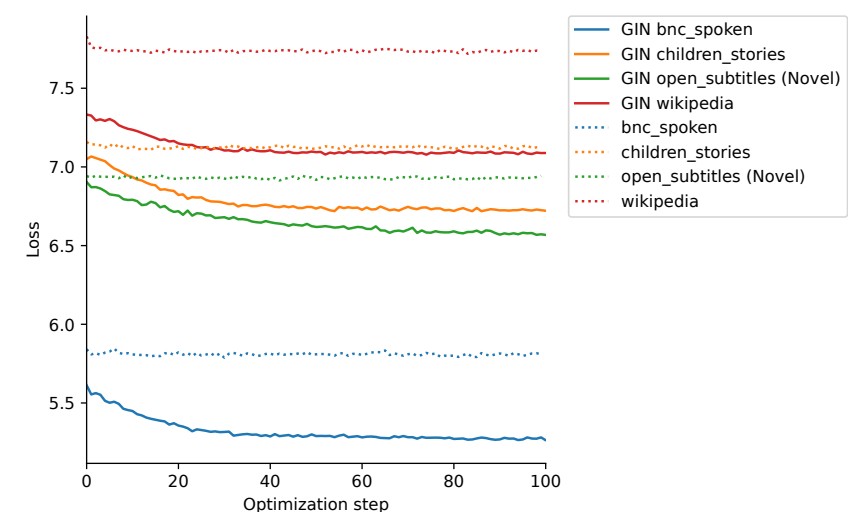

Figure 15: An example run showing the loss on predicting words from the same sequence while optimizing the latent context inputs

# E   LANGUAGE MODEL

## E.1   FURTHER EXPERIMENTAL DETAILS ON TRAINING TASK-AWARE LANGUAGE MODEL

We based our language modeling approach on the official PyTorch example for training a word-level language model, which can be found at `https://github.com/pytorch/examples/tree/main/word_language_model`. We largely adopted the original design choices and parameters, including the usage of a stochastic gradient descent optimizer with a decaying learning rate, a sequence length of 35 words and a batch size of 20, gradients clipped to a value of 0.25, input layer and output layer dropout rate 0.25, word-level tokenizer, and a log-softmax function at the final output layer. We used the LSTM implementation. The tokenizer's output was passed through an embedding layer, followed by concatenation with a latent representation, specifically a one-hot encoding of the dataset ID. The LSTM output passed through an MLP layer to project the LSTM's hidden units back to the number of tokens, and a log-softmax function was applied as the final output activation.

Note that we trained the model on 3 datasets at a time and tested generalization on a fourth dataset to keep the tokenizer from indexing more than 1 million distinct words. Adding more datasets lead to the embedding layer having a massive size that would not fit in a GPU memory.

Here we provide brief information on the datasets included in the BaybLM challenge. We use their more challenging smaller "Strict-Small" data.

| Dataset | Domain | Dataset Size |
|---|---|---|
| AoCHILDES (MacWhinney, 2000) | Child-directed speech | 0.44M |
| British National Corpus (BNC),[1] dialogue portion | Dialogue | 0.86M |
| Children's Book Test (Hill et al., 2016) | Children's books | 0.57M |
| Children's Stories Text Corpus[2] | Children's books | 0.34M |
| Standardized Project Gutenberg Corpus (Lahiri, 2014) | Written English | 0.99M |
| OpenSubtitles (Lison & Tiedemann, 2016) | Movie subtitles | 3.09M |
| QCRI Educational Domain Corpus (QED; (Abdelali et al., 2014)) | Educational video subtitles | 1.04M |
| Wikipedia[3] | Wikipedia (English) | 0.99M |
| Simple Wikipedia[4] | Wikipedia (Simple English) | 1.52M |
| Switchboard Dialog Act Corpus (Godfrey et al., 1992) | Dialogue | 0.12M |
| *Total* | – | 9.96M |

Table 7: The datasets included in the BabyLM Challenge (Warstadt et al., 2023), please see the original paper for further details on datasets and sources. The authors reported the number of words in the training set of each corpus. [1]`http://www.natcorp.ox.ac.uk` [2]`https://www.kaggle.com/datasets/edenbd/children-stories-text-corpus` [3]`https://dumps.wikimedia.org/enwiki/20221220/` [4]`https://dumps.wikimedia.org/simplewiki/20221201/`

