# OpenReview forum: "Gradient-based inference of abstract task representations for generalization in neural networks"
_ICLR.cc/2025/Conference — Submitted to ICLR 2025_

### Official Review · Reviewer_vL85 · 2024-10-29

**Soundness:** 2
**Presentation:** 3
**Contribution:** 2
**Rating:** 5
**Confidence:** 3

**Summary:**

The authors propose a method called GBI to train a recurrent and convolutional neural network to infer the task category during test time. The inference is driven by gradient updates only at the task category layer, which can either be done iteratively or by approximating the maximal entropy point, while keeping the rest of the weights fixed. The authors demonstrated the benefits of this method across a variety of toy, image generation and language generation tasks in terms of a lower training loss. Additionally, the authors argue that once the model learns different task representations, only the task category layer needs to be optimized to decrease test or generalization error.

**Strengths:**

-- included anonymous code for replication
-- demonstrated generality of claim that using gradients to infer task category improves performance across 3 modalities, toy, vision and language tasks.
-- One step gradient update by symbolically estimating the optimal point seems to be novel.
-- Gradient based inference improves interpretability during training (e.g. bayesian inference like estimation) and to identify OOD samples.

**Weaknesses:**

-- Please include a diagram of the GBI-LSTM architecture for the toy and language tasks. Specifically which synapses are modified during training and inference time i.e. which is the task abstraction layer/weights z? (Pg 5, line 242)
-- The idea to optimize only the input representation weights instead of the entire model is not novel. This idea dates long back to the idea of learning schemas and adjusting new information to fit the prior learned template (Lampinen, McClelland 2020 PNAS; Kumar et al. 2024 arXiv 2106.03580).
-- the claim that GBI-LSTM shows no signs of forgetting compared to the LSTM is not strong. The baseline MSE performance is 0.24, which the GBI maintains for new datasets, but the deviation by LSTM does not seem to be significant (Table 1). Was a statistical test done to compare LSTM and GBI performance?
-- Why learning to infer task category improves learning and generalization in these tasks is unclear. Perhaps the authors can perform low dimensional analysis to show how the network learns to represent different datasets into non-overlapping subspaces and during inference, the network's activity converges towards a specific prior learned subspace or learns to compose them (Lin et al. 2024 arXiv 2309.04504)?
-- The authors argue that GBI improves generalization loss in language prediction task. Although the baseline LSTM shows consistent loss of 6.8 (I assume all model weights are fixed), the GBI loss starts off higher of around 6.95 and decreases to 6.6 over 100 optimization steps. Does the loss continue to decrease with longer optimization steps? If not, is a generalization loss of 6.6 significant compared to 6.8?
-- Given that the models are an LSTM and not a large model, I thin it is reasonable to expect at least 30 seed runs instead of 4 as in Fig. 5D to increase the confidence in results, especially when the difference afforded by LSTM and GBI is small.
-- Since the authors used LSTM for toy dataset, and a CNN for image, it would have been a solid contribution if the authors demonstrated GBI using a simple transformer architecture for language prediction instead of LSTM.

**Questions:**

-- Is the one-step gradient update method novel? Or was it developed prior?
-- giving task category as input should significantly reduce the training complexity of needing to infer the task (Kumar et al. 2022 Cerebral Cortex). Why was the difference in training loss not as apparent? Did the authors perform hyper parameter sweeps for the learning rate and number of units? It is easy to choose a set of hyper parameters where the distinction between LSTM and GBI is artificially similar.
-- Why was LSTM chosen instead of GRU or a Vanilla RNN? Training an LSTM on the simple toy dataset might be an overkill.
-- why is the ratio of shared units between LSTM and GBI different in the 0th binned training block (Fig. 2C)? They should aggregated such that the initialized activation is the same for comparison.

---

> ### Author Response · Authors · 2024-11-25
> **Response to reviewer vL85 (1/3 comments)**
>
> >Gradient based inference improves interpretability during training
>
> We thank the reviewer for this comment. The reviewer notes that our framework may have an additional benefit we have not considered. Our method involves neural networks inferring a low-dim task abstraction prior to performing a task, observing what low-dimensional task abstraction the network infers may indeed contribute to the interpretability of neural computations.
>
> >Please include a diagram of the GBI-LSTM architecture for the toy and language tasks. Specifically which synapses are modified during training and inference time i.e. which is the task abstraction layer/weights z? (Pg 5, line 242)
>
> We agree. A diagram would more visually and clearly define what is being optimized. We now include this in Fig 3 as the first subpanel. Specifically, Z is a vector of units that feeds as input to the neural network through a with a standard set of weights (i.e., a weight matrix with dimensions [dim(Z), neural network hidden units size]). Importantly, the projection from Z to the network is optimized during training as part of the neural network parameters. During inference we only optimize the activations of the Z units (i.e., their neural activation (firing rates)).
>
> > The idea to optimize only the input representation weights instead of the entire model is not novel. This idea dates long back to the idea of learning schemas and adjusting new information to fit the prior learned template (Lampinen, McClelland 2020 PNAS; Kumar et al. 2024 arXiv 2106.03580).
>
> Thank you for pointing this out. Upon reflection, we recognize that our previous descriptions may have led to a misunderstanding. Unlike the methods cited, which optimize input representation weights or low-rank perturbations to adapt to new tasks while treating the neural network as a fixed reservoir, our approach differs in key ways. Specifically, we do not optimize the task abstraction input weights to the network. Instead, we optimize the low-dimensional activations of the Z units (e.g. 2 units in the toy task, and 10 units in image generation experiments). We focus on understanding how neural networks can be trained to respond to low-dimensional task abstractions, and then infer them by optimizing the task abstraction units directly during testing. This distinction sets our work apart from prior methods that rely on gain modulation or low-rank updates to preserve prior knowledge.
>
> > giving task category as input should significantly reduce the training complexity of needing to infer the task (Kumar et al. 2022 Cerebral Cortex).
>
> We found Kumar et al. 2022 and 2024 to be very relevant work with similar motivation to disentangle learning  a computation from forming a representation of the computation itself. We now cite these works as models with a distinct task representation layer. Such methods are expected to reduce the complexity of training and create adaptable models that can be adapted simply by changing their task representation input.
>
> >Why was the difference in training loss not as apparent? Did the authors perform hyper parameter sweeps for the learning rate and number of units? It is easy to choose a set of hyper parameters where the distinction between LSTM and GBI is artificially similar.
>
> We agree with these observations. The difference in training is modest in our case because the tasks are of high complexity (e.g. learning wikipedia dataset) while we provide coarse task abstractions (e.g. dataset identifier). We expect that models that discover task abstractions from data to have more complex task abstractions that further break down the computation space and lead to larger differences in training loss. There have been a few examples of models that do this of late (Hummos, ICLR 2023; Butz et al. 2019, Sandbrink et al., NeurIPS 2024).
>
> Hummos, A. Thalamus: a brain-inspired algorithm for biologically-plausible continual learning and disentangled representations. (ICLR, 2023).
>
> Butz et al., Learning, planning, and control in a monolithic neural event inference architecture. Neural Networks 117, 135–144 (2019).
>
> Sandbrink et al., Neural networks with fast and bounded units learn flexible task abstractions. (NeurIPS 2024, spotlight)

---

> > ### Author Response · Authors · 2024-11-25
> > **Response to reviewer vL85 (2/3 comments)**
> >
> > > the claim that GBI-LSTM shows no signs of forgetting compared to the LSTM is not strong. The baseline MSE performance is 0.24, which the GBI maintains for new datasets, but the deviation by LSTM does not seem to be significant (Table 1). Was a statistical test done to compare LSTM and GBI performance?
> >
> > In response to this comment, we noticed that our previous Table 1 did not offer a clear picture of the comparisons done. We removed the first row which the reviewer cites its values. To clarify, the small difference between LSTM and GBI-LSTM in the removed row is because it measured performance on data points encountered at the very end of training. In this scenario, neither network shows forgetting, as both are evaluated on recently trained tasks. For example, in a training curriculum like [Task A, Task B, Task A], this row tested performance on Task A, which does not demonstrate forgetting. The key difference of interest is in performance on earlier tasks, such as Task B, where forgetting is more relevant.
> >
> > By removing this control test, the revised table better represents the findings. Thank you for pointing this out.
> >
> >  > Why learning to infer task category improves learning and generalization in these tasks is unclear. Perhaps the authors can perform low dimensional analysis to show how the network learns to represent different datasets into non-overlapping subspaces and during inference, the network's activity converges towards a specific prior learned subspace or learns to compose them (Lin et al. 2024 arXiv 2309.04504)?
> >
> > While studying these mechanisms would be greatly interesting, several recent papers, including the one the reviewer cites, detailed accounts of how generalization happens by reusing shared computational motifs inside the neural networks (Yang et al. 2019, Driscol 2024, Goudar et al. 2023). We do not believe we will be able to go beyond what these papers found.
> >
> > Yang, G. R., Joglekar, M. R., Song, H. F., Newsome, W. T. & Wang, X.-J. Task representations in neural networks trained to perform many cognitive tasks. Nat Neurosci 22, 297–306 (2019).
> >
> > Goudar, V., Peysakhovich, B., Freedman, D. J., Buffalo, E. A. & Wang, X.-J. Schema formation in a neural population subspace underlies learning-to-learn in flexible sensorimotor problem-solving. Nat Neurosci 26, 879–890 (2023).
> >
> > Driscoll, L. N., Shenoy, K. & Sussillo, D. Flexible multitask computation in recurrent networks utilizes shared dynamical motifs. Nat Neurosci 27, 1349–1363 (2024).
> >
> > > The authors argue that GBI improves generalization loss in language prediction task. Although the baseline LSTM shows consistent loss of 6.8 (I assume all model weights are fixed), the GBI loss starts off higher of around 6.95 and decreases to 6.6 over 100 optimization steps. Does the loss continue to decrease with longer optimization steps? If not, is a generalization loss of 6.6 significant compared to 6.8?
> >
> > Thank you for the insightful observation. The generalization loss does continue to trend downward with more optimization steps, but we do not expect a significant further decrease. This aligns with the coarse nature of the task abstractions used (e.g., dataset identifiers) compared to the complexity of the tasks (e.g., Wikipedia data). Our goal at this stage is to make a conceptual point, demonstrating the qualitative flexibility of models with task abstractions. We anticipate larger improvements in generalization when task abstractions are richer, such as those reflecting topics, paragraph intent, or sentence embeddings. Ultimately, we envision that models capable of discovering such abstractions from data will yield even greater benefits (Hummos, ICLR 2023; Butz et al. 2019, Sandbrink et al., NeurIPS 2024).
> >
> > > Given that the models are an LSTM and not a large model, I thin it is reasonable to expect at least 30 seed runs instead of 4 as in Fig. 5D to increase the confidence in results, especially when the difference afforded by LSTM and GBI is small.
> >
> > We share the reviewer’s intuition. Our initial submission did not accurately describe what we did. We trained each model for 4 seeds, but with 12 different choices for the three training datasets (12 sets of 3 datasets). Each of the 12 training sets was then trained over 4 seeds or random initializations of the network, leading up to results aggregated from 48 model runs. We now correctly describe this experiment structure in the main text figure 5 legend. Thanks for pointing this out.

---

> > > ### Author Response · Authors · 2024-11-25
> > > **Response to reviewer vL85 (3/3 comments)**
> > >
> > > > Since the authors used LSTM for toy dataset, and a CNN for image, it would have been a solid contribution if the authors demonstrated GBI using a simple transformer architecture for language prediction instead of LSTM.
> > >
> > > Very good suggestion. We in fact have ongoing work applying the framework to transformers, but we never connected that to this paper. Our initial work with transformers however revealed that this is surprisingly quite involved. Briefly, transformers have discrete tokens as input, so this would not necessarily support gradient based inference that finds linear combinations of task abstractions to generalize. (i.e., combining two tokens will likely be nonsensical). Additionally, the transformers architecture has many sub-networks, and choosing where to input task abstractions has also required significant exploration. Currently, we are using a LoRA scheme where task abstractions are the low rank representations. We might update this paper prior to the camera-ready if we have simple results that are working well, but we cannot promise that these would be ready in time. Thanks for this thought.
> > >
> > >
> > > > Is the one-step gradient update method novel? Or was it developed prior?
> > >
> > > We are aware of one study that shows that the first few gradient updates are quite informative, but they use the gradients as input to a separate network that then updates the latent. Marino et al, 2018.
> > > Marino, J., Yue, Y. & Mandt, S. Iterative Amortized Inference. Proceedings of Machine Learning Research 80, 3403–3412 (2018).
> > >
> > > > Why was LSTM chosen instead of GRU or a Vanilla RNN? Training an LSTM on the simple toy dataset might be an overkill.
> > >
> > > Thank you for raising this interesting point. Our thinking behind choosing an LSTM is to use a baseline model with multiple gated interactions built-in. We see that providing task abstractions during training produces task modules, and the task abstractions, in a way, gate in or out the appropriate task modules. By choosing an LSTM, we now know that even with gated interactions, such explicit gating to handle multi-tasks does not emerge from the standard ML training paradigm, even if the capacity for it exists in the model.
> > >
> > > > why is the ratio of shared units between LSTM and GBI different in the 0th binned training block (Fig. 2C)? They should aggregated such that the initialized activation is the same for comparison.
> > >
> > > Thanks for pointing this out. We will review our binning strategy. There was a minimum number of blocks we had to aggregate data from to get a stable estimate, but it might be possible to test the model while it is frozen, after each training block, rather than use the LSTM responses during training to run the analysis. We will update this soon.

---

> ### Comment · Reviewer_vL85 · 2024-11-26
> **Maintain score**
>
> I appreciate the authors' efforts in addressing my concerns. Nevertheless, I believe it is necessary to maintain the current score. I recommend conducting additional experiments with various architectures to determine if there are significant performance improvements. Moreover, the submission would benefit from analysis and ablation studies on representational motifs to better understand their impact on generalization.

---

### Official Review · Reviewer_4ta1 · 2024-10-29

**Soundness:** 3
**Presentation:** 3
**Contribution:** 3
**Rating:** 6
**Confidence:** 3

**Summary:**

Inspired by cognitive science, the authors identify the lack of an efficient module for inferring the current task in current models. According to the authors, previous approaches to task inference do not meet the following two assumptions:
1. lack of efficient detection if the task has been repeated
2. lack of recomposing mechanism of previously learned tasks.
The authors propose an approach grounded in variational inference, using an expectation-maximization-like framework, and they demonstrate GBI’s effectiveness across synthetic, image classification, and language modeling tasks.

**Strengths:**

1. Clear and relevant objective. The aim of the work is clearly defined.
2. The authors provide code.
3. Experimental validation on various scenarios, from synthetic datasets to complex tasks (image classification and language modeling).

**Weaknesses:**

Although I appreciate that the authors examined their method on various scenarios, I am not sure if the complexity of these tasks is enough. For example, in OOD detection, adding a comparison of CIFAR10 vs SVHN datasets would be valuable.

**Questions:**

I am very curious about catastrophic forgetting matters. The authors gently mentioned this feature of their method, but it was only evaluated on the toy dataset; why? Could you provide more experiments in this area?

---

> ### Author Response · Authors · 2024-11-25
> **Response to reviewer 4ta1 (1/2 comments)**
>
> >Clear and relevant objective. The aim of the work is clearly defined.
>
> Thanks for the encouraging remarks.
>
> >Although I appreciate that the authors examined their method on various scenarios, I am not sure if the complexity of these tasks is enough.
>
> Thank you for highlighting this important point regarding task complexity. We agree that the tasks examined in our study are relatively simple, and the reviewer's concern about scaling to larger vision or language datasets is valid. Our exploration of scaling to the CIFAR-100 dataset indeed revealed valuable insights about the requirements for meaningful scaling.
>
> In what follows we expand on two main points. A promising class of models have emerged of late with neural models capable of forming their own task abstractions directly from data. First, we offer a foundational contribution to these models by exploring the properties of gradient-based inference, which they heavily use. Second, for meaningful scaling up of our results, future work will have to rely on those models to provide richer task abstractions, beyond the human-provided labels we use in this work.
>
> A promising direction is an emerging class of models that relies on gradient-based Expectations Maximization dynamics to identify tasks in their training data and label them with internally generated task abstractions  (Hummos, ICLR 2023; Butz et al. 2019, Sandbrink et al., NeurIPS 2024). These models simply optimize \theta (the neural network parameters) and $Z$ (the task abstraction layer) through gradient descent, with $Z$ having a faster learning rate. This straightforward setup can dynamically form task abstractions in $Z$, allowing the network parameters to organize into modules specific to each task. Such models have demonstrated advantages in mitigating catastrophic forgetting, enhancing adaptability, and improving generalization.
>
> Hummos, A. Thalamus: a brain-inspired algorithm for biologically-plausible continual learning and disentangled representations. (ICLR, 2023).
>
> Butz et al., Learning, planning, and control in a monolithic neural event inference architecture. Neural Networks 117, 135–144 (2019).
>
> Sandbrink et al., Neural networks with fast and bounded units learn flexible task abstractions. (NeurIPS 2024, spotlight)
>
> However, these internally generated task abstractions pose challenges for evaluation as they can drift significantly during training. It becomes difficult to assess the accuracy of gradient descent as an inference mechanism—how effectively can it retrieve previously learned tasks, handle uncertainty in the $Z$ space, or detect out-of-distribution data?
>
> Our study addresses the limitation by using human-provided labels as task abstractions, such as image class in the image generation experiment. The gradient-based EM methods use iterative optimization with hundreds of optimization passes, while we here found that one-step gradients might be sufficient. In addition, we provide estimates of how accurate gradient based inference might be.
>
> Our first point here is: this work offers a foundation for gradient-based EM methods. In fact, one of those papers already cited an earlier version of this work, though we will not specify further to maintain anonymity.
>
> However, our efforts to scale up to CIFAR-100 revealed the limitations of these human-provided labels. In this case, the model was given the image class as a task abstraction to guide image reconstruction. Despite this, backpropagation largely ignored the additional task information and relied more heavily on visual features from the encoder. This suggests that image class labels do not significantly reduce variance in the pixel space, limiting their utility to the model. This is reflected in low accuracy for GBI in inferring image class. We concluded that low-complexity, human-provided task abstractions are insufficient for describing more complex datasets.
>
> Our second point, to meaningfully scale up the framework in this work will have to rely on gradient-based EM methods for richer task abstractions.
>
> Reflecting on this, we recognize that our original manuscript could have framed these motivations more clearly, which would better highlight the unique contributions of this work. We now describe these considerations in the introduction to motivate the study of gradient based inference. We also add additional discussion as we introduce CIFAR100 results and explain what it will take to scale up.
>
> We hope that the possibility of a contribution to answer foundational questions for this novel class of models might offset the limited complexity of the datasets this work tackles.

---

> > ### Author Response · Authors · 2024-11-25
> > **Response to reviewer 4ta1 (2/2 comments)**
> >
> > >For example, in OOD detection, adding a comparison of CIFAR10 vs SVHN datasets would be valuable.
> >
> > Based on the above considerations, GBI network trained on CIFAR 10 largely ignores class information as it adds little to reduce the variance in images. As such, we already expect that it would not do as well as identifying those images from an OOD dataset.
> >
> > Nonetheless, we will make another attempt to address the accuracy by forcing the model further to use the task abstractions, even if not fully informative of the image. Should we succeed, we will post an update to the forum here, or might include the results in the camera-ready version, should the paper be accepted.
> >
> > Additionally, to more immediately address the reviewer’s concerns we now, first, tone down our claims about OOD detection as specific to MNIST and fMNIST datasets, and, second, we add a sentence indicating why we did not pursue OOD experiments on CIFAR datasets.
> >
> >
> > >I am very curious about catastrophic forgetting matters. The authors gently mentioned this feature of their method, but it was only evaluated on the toy dataset; why? Could you provide more experiments in this area?
> >
> > Thank you for raising this point. Training models with task abstractions provided allows the neural network to form task modules which alleviates forgetting. However, this current work assumes access to task boundaries and task IDs, which are strong assumptions for a continual learning method. As such we chose to not engage with continual learning benchmarks in this work. We again refer to models that generate their own task labels from data. These models can detect task boundaries and task IDs with no supervision. Recent work showed that this framework does indeed produce practical solutions to continual learning on simple benchmarks (Hummos, 2023).

---

> > > ### Comment · Reviewer_4ta1 · 2024-11-26
> > >
> > > Thank you for your detailed responses to my comments and for addressing the concerns raised in my initial review. I appreciate your acknowledgment of your work's limitations and efforts to clarify your contributions' real value and potential impact.
> > >
> > > Your exploration of gradient-based inference and the foundational insights provided into emerging neural models that autonomously generate task abstractions are significant. While I commend these contributions and recognize the thoughtfulness of your revisions, I believe that certain aspects—such as the challenges of applying your method to more complex datasets and the limited experimental scope regarding issues like OOD detection and catastrophic forgetting—deserve deeper investigation.
> > >
> > > Therefore, I finalize my recommendation as borderline accept. I believe your work is valuable and worthy of inclusion, but further development in future iterations could enhance its impact.

---

### Official Review · Reviewer_MWGp · 2024-11-03

**Soundness:** 2
**Presentation:** 2
**Contribution:** 2
**Rating:** 5
**Confidence:** 3

**Summary:**

This paper proposes a method of incorporating a task obstruction process to improve the performance of neural networks. It considers the framework in which a likelihood function is maximized with respect to the network weights with task abstraction data included as inputs of contextual information in the training phase and the abstraction data is estimated via gradient descent in the test phase. It proposes a method of approximating the iterative optimization of task obstruction with a one-step gradient update. The proposal is based on insights from neuroscience about the roles played by intermediate abstraction in performing tasks and techniques in variational inference to handle contextual information with latent variables.

**Strengths:**

- A simple and effective method of approximating the iterative optimization of task obstruction is proposed. It is achieved by taking the maximum entropy point as the initial value and updating it with a one-step gradient update. The validity of this approximation is checked experimentally (Fig. 3G, H).

- Experiments show the superiority of the proposed method to the canonical classifier in OOD detection(Table 3, Fig 4A, B, C).

**Weaknesses:**

- The training methods used in the experiments (sequence prediction by an RNN and data reconstruction by an autoencoder with its latent variables concatenated with contextual information) are not exactly the same as the one mentioned in Section 2 (MLE of the likelihood function (1)). It is not mentioned in the paper how the extensions to those variants do (or do not) affect the argument regarding the proposal in Section 2.

- The purpose of the experiments comparing LSTM and GBI-LSTM (Fig. 2, Fg.3D,E, 5A,B,D) is not clear. It seems to me that they are just confirming the impact of the input of task labels for the sequence prediction problems with multiple tasks. It is preferable if this point is clarified in line with the motivations or expectations in Section 1 or Section 2.

- Experiments imply that the canonical classifier may have better accuracy than the proposed methods (Table 2), but no reasoning for this is provided. It does not throw away the value of the superiority in OOD detection. Rather, the trade-off relation between them is worth a remark if it is confirmed to exist.

- The following references are not found in the submission: (L357)"supplementary material". (L396)"Table S8"

**Questions:**

- Please consider the possibility of the clarifications mentioned in the weaknesses section.

---

> ### Author Response · Authors · 2024-11-25
> **Response to MWGp (1/2 comments)**
>
> >The training methods used in the experiments (sequence prediction by an RNN and data reconstruction by an autoencoder with its latent variables concatenated with contextual information) are not exactly the same as the one mentioned in Section 2 (MLE of the likelihood function (1)). It is not mentioned in the paper how the extensions to those variants do (or do not) affect the argument regarding the proposal in Section 2.
>
> We appreciate the reviewer for raising this important conceptual point. This took a bit of reflection. To ensure we are aligned on the interpretation of the differences between the theoretical methods and implemented experiments, we welcome further clarification if needed.
>
> Our understanding is as follows: the methods section primarily considers a likelihood function of the form L=f(X,Z). In contrast, the RNN and autoencoder experiments incorporate additional inputs into this computational graph. Specifically, the RNN uses hidden states evolving from previous inputs, while the autoencoder leverages latent encodings produced by the encoder. Notably, these additional inputs are themselves trainable, potentially complicating the theoretical analysis.
>
> The reviewer has rightly highlighted the need to consider how these extensions might influence the behavior of the model compared to the theoretically motivated framework. One way to reconcile this is to view the encoder and RNN as transformations of the input X, effectively modifying its distribution. While this induces non-stationarity in the distribution of X during training, evidence (e.g., Wu et al., 2020 https://arxiv.org/abs/2004.09189) suggests that the encoder often converges faster than the decoder. This faster convergence could reduce the transient non-stationarity from the perspective of the decoder, rendering the extensions to the computational graph less impactful on the overall argument.
>
> Another way to reconcile is to consider the latent variables from the encoder, and the hidden units activations from the RNN, as part of the model parameters. Our framework calls for updating the parameters of the model during training, and we can show that doing so, also updates the encoder latent and RNN hidden activations according to their gradients. Thus we can treat them as part of the model parameters being optimized during training, and our theoretical description of the methods should hold.
> To briefly state this symbolically for the RNN case:
>
> The hidden state of the RNN is updated as $h_t = W h_{t-1}$. During training, the weights $W$ are updated via gradient descent:
> \begin{equation}
> \Delta W = -\eta \frac{\partial L}{\partial W} = -\eta \frac{\partial L}{\partial h_t} (h_{t-1})^T
> \end{equation}
> where $\eta$ is the learning rate. After the weight update, the new hidden state is given by:
> \begin{equation}
> h_t^{\text{new}} = (W + \Delta W) h_{t-1} = h_t - \eta \frac{\partial L}{\partial h_t} \|h_{t-1}\|_2^2
> \end{equation}
>
> with $$\|h_{t-1}\|_2^2 = h_{t-1}^T h_{t-1}$$
>
> This shows that updating the weights $W$ during training indirectly updates the hidden state $h_t$ as well, effectively performing a gradient descent step on $h_t$ with respect to its gradient $\frac{\partial L}{\partial h_t}$, scaled by $\|h_{t-1}\|_2^2$. This aligns with our framework, where we treat the RNN hidden activations $h_t$ as part of the model parameters being optimized during training.
>
> We are still writing a brief overview of these considerations to add to the paper. Thanks for the helpful comment.

---

> > ### Author Response · Authors · 2024-11-25
> > **Response to MWGp (2/2 comments)**
> >
> > >The purpose of the experiments comparing LSTM and GBI-LSTM (Fig. 2, Fg.3D,E, 5A,B,D) is not clear. It seems to me that they are just confirming the impact of the input of task labels for the sequence prediction problems with multiple tasks. It is preferable if this point is clarified in line with the motivations or expectations in Section 1 or Section 2.
> >
> > Thank you for raising this important point. We added two new subpanels to Fig 2 to concretely show the points made. We also have revised the paper to better articulate the purpose of the experiments comparing LSTM and GBI-LSTM and their connection to the framework and claims in Sections 1 and 2.
> >
> > We first group the points we wish to make into two conceptual categories: 1) the benefits of **training** with task abstractions provided: faster learning and reduced forgetting. 2) The benefits during **testing**, when task abstraction values are no longer available, and we rather use gradient-based inference (GBI) to infer them: accurate task inference using one-step gradients, generalization by recomposing task abstractions, and finally OOD detection.
> >
> > We show a subset of these features in each of the experiments training models on data from three different domains. Each of the domains enabled us to make unique points, not possible or not as meaningful in the others, in addition to showcase that the method is domain-general.
> >
> > Fig 2 showed the benefits of training with task abstractions on a toy dataset. We added two new panels A and B to this figure, to concretely show how training the LSTM with task abstractions is qualitatively different. Forgetting here was particularly meaningful because the task is exceedingly simple, and we throw a 100 unit LSTM which has the capacity for internal gating to emerge, yet still suffers from forgetting. Fig 3D, E show the benefits of using gradients to infer task abstractions during testing, showing that GBI-LSTM generalizes better. Fig 5 A, B, show, in a language model, one effect of task abstraction during training: faster learning. Fig 5 D, shows in the language model the effect of GBI during testing: generalization to novel datasets. Generalization here is more meaningful than the toy task, because the tasks are of more interesting level of complexity (language datasets). Finally, the claims of accuracy of one-step gradients for task inference, and OOD detection, were only possible to show in the image generation experiment, because it had many possible values for task abstractions (10 in MNIST and 100 in CIFAR100), and had well established methods and experiments to test OOD detection.
> >
> > To make this clearer in the revised manuscript, we have:
> > 1. **Conceptually grouped the points to be made**: Grouped the benefits of the framework into the above two categories: the benefits of task abstractions during training, and the benefits of using GBI to infer them during testing. This distinction now guides the presentation of all experiments.
> > 2. **Improved Section Titles**: Updated section titles to explicitly highlight the findings and align with the claims in the introduction.
> > 3. **Added Context and Summaries**: Each experimental section now begins with a clear statement of its purpose tying the results back to the broader framework.
> >
> > Thank you for this valuable comment, we believe that the revised manuscript benefited greatly from this conceptual structure..
> >
> > >The following references are not found in the submission: (L357)"supplementary material". (L396)"Table S8"
> >
> > Implementation details for likelihood regret were omitted. We apologize. Added now to appendix D.
> >
> > Table S8 exists, but the sentence in the manuscript does not state what to expect in the table. It reports CIFAR100 results without the multiplicative effect that we added to improve results (i.e., task abstractions Z are fed as an additive input, as opposed to projected to a gating mask that is applied multiplicatively). This is a technical detail we grappled with, and we now decided to move it entirely to the supplementary for clarity. Thank you.
> >
> > >Experiments imply that the canonical classifier may have better accuracy
> >
> > Thank you for identifying this ambiguity in our presentation. The drop in accuracy with the proposed method (GBI) is expected compared to a classifier explicitly trained for accurate classification. Our aim is not to achieve the highest accuracy but to ensure the drop remains manageable, which our results confirm. GBI maintains reasonable accuracy while offering additional benefits detailed in the paper.
> >
> > In our revised manuscript, we clarified this point and replaced vague references to the classifier as “canonical methods.” We also elaborated on the motivation for comparing GBI to a classifier to provide a clearer context for the results.
> >
> > We thank the reviewer again for their time, effort and the thoughtful feedback. We hope our responses and changes to the manuscript addressed the comments raised.

---

> > > ### Comment · Reviewer_MWGp · 2024-12-03
> > >
> > > Thank you for your detailed response and for working on clarifications. Now, the claims of the paper are clear to me. The theoretical expectations about the extension to RNN and the autoencoder model are interesting. However, the argument currently provided is too intuitive and has not ensured that the empirical facts found in Section 3 serve as checks of the proposed method. I decide to keep the score.

---

### Official Review · Reviewer_v6Yp · 2024-11-04

**Soundness:** 2
**Presentation:** 2
**Contribution:** 3
**Rating:** 5
**Confidence:** 3

**Summary:**

The authors introduce Gradient-Based Inference (GBI) of abstract task representations, a method that enables neural networks to infer and adapt task representations dynamically, promoting faster learning and better generalisation. Inspired by human adaptability—where task abstractions allow flexible responses to the same input depending on internal goals—their approach enables neural networks to infer and adapt task representations on the fly.

They frame the setting as an optimisation problem through variational inference, and their GBI uses backpropagated gradients to infer and adjust task representations in a neural network. Experiments in a range of domains including image classification and language modelling demonstrate benefits in learning efficiency, generalisation, and reduced forgetting, as well as its performance in uncertainty estimation and out-of-distribution detection.

**Strengths:**

- originality and contributions: This approach brings insights from cognitive science on human task learning and generalisation to deep learning. Their gradient-based inference (GBI) method introduced in the paper is novel, and seems to be an innovative application of gradients in task inference and adaptation beyond traditional optimization. Furthermore, by positioning GBI as a model capable of estimating uncertainty and detecting out-of-distribution samples, the work brings a fresh perspective and could potentially bring new insights to the field.
 - significance: the problem setting is an important one: enabling artificial agents to flexibly to situations depending on their varying goals. The connections made to human and animal cognition and learning provide a solid foundation for this setting, grounding the approach in well-established principles of adaptive behaviour and task representation in human learning.
 - The paper demonstrates the effectiveness across varied tasks highlighting its potential as a versatile domain-agnostic approach. Overall their results demonstrate some promising advantages in learning efficiency, generalisation, and uncertainty estimation.
 - The inclusion of the code (via link to an anonymous repo) is appreciated for reproducibility and clarifying implementation details (however the repository would benefit from better organisation, see weaknesses).

**Weaknesses:**

- motivation for gradient-based approach: the motivation for adopting a gradient-based approach could be clearer, as the paper does not sufficiently explain why gradients offer an advantage for task inference and recomposition over alternative methods. While gradient-based updates are obviously often employed in optimisation, it is less intuitive why they would be particularly effective in inferring abstract task representations or detecting out-of-distribution samples. A more thorough discussion grounding and comparing the gradient-based approach with other task inference methods, and meta-learning perspectives could help to clarify the benefits here.
 - The section on the one-step gradient update and maximal entropy initialisation could benefit from a clearer, more intuitive explanation. To improve clarity, the authors could add a visual schematic that illustrates the process step-by-step, and how a single gradient update shifts this initial state towards a more task-specific representation.
 - Improving coherence: the paper would benefit from better coherence as the flow between sections feels disjointed. This makes it challenging to follow the core narrative. While each section presents important concepts mostly clearly, there is often a lack of clear transitions that tie ideas together leaving the reader to infer the connections. For example, the jump from theoretical discussions of gradient-based inference to experimental details seems abrupt, and then later suffers from limited explanation of how each experiment directly relates back to the proposed framework. A bit more introduction and summarisation at the beginning and end of sections, focused on tying each section to the core ideas would improve the flow of the paper.
 - limited use of intuitive examples: incorporating a few straightforward examples or analogies could provide readers with a more accessible understanding of the contributions being presented. For instance, using a simple scenario (like inferring a task from partial information in an everyday context) could illustrate how the model operates based on gradient information to infer task information and provide useful uncertainty estimates.
 - lack of comparison to important uncertainty estimation methods: while the paper claims advantages in OOD detection and uncertainty estimation, the results provided are not compared against established methods like ensemble approaches.
 - poorly organised code: to improve accessibility, the authors could streamline the repository, making it quicker and easier to connect model implementations with the methods and experiments described in the paper.

**Questions:**

- can the motivation for the gradient-based inference be described more clearly and intuitively? particularly, relating back to the problem setting and the limitations of existing methods.
 - how do the uncertainty estimates and OOD detection capabilities provided by the model compare to other approaches like ensembles or Bayesian neural networks?
 - given that the authors ground the problem setting and draw inspiration from human cognition and learning, do the authors feel their approach has biological plausibility at some level?

Overall, the discussed limitations hold it back from a higher score, however, the paper's originality and promising empirical results make it a interesting contribution. With a clearer exposition and better grounding/comparisons to other approaches, I would raise my score.

---

> ### Author Response · Authors · 2024-11-25
> **Response to v6Yp (1/2 comments)**
>
> >the work brings a fresh perspective and could potentially bring new insights to the field.
>
> We thank the reviewer for the encouraging comments.
>
> The reviewer made several well thought out suggestions to improve the paper, leading to several changes and additions as we detail below.
>
> >motivation for gradient-based approach
>
> Thank you for this comment. We agree that our initial submission did not motivate the advantages of using gradients for task inference. Gradients provide two key benefits. First, is avoiding the alignment problem: task abstractions in neural systems influence computations via the parameters they modulate. In other methods, inferred task abstractions can become misaligned as network parameters are updated during training. In contrast, gradients are computed based on the current state of the network, ensuring alignment between the task abstractions and the underlying computations. We have clarified this point in the introduction to better motivate the use of gradients as a solution.
>
> Second, a novel set of models appeared recently that train models by both updating the weights, and also updating a task abstraction layer with a higher learning rate (Hummos, ICLR 2023; Butz et al. 2019, Sandbrink et al., NeurIPS 2024). This simple setup can surprisingly discover tasks in the unlabeled stream of data, represent them as distinct units, and switch task abstractions appropriately to solve previous tasks and compose solutions to new ones. These models highlighted the need for a principled study of inference through gradients, but because the task abstractions are internally generated, and drift during training, it is difficult to assess how well gradient-based inference works in this setting. Our paper, instead, uses human provided labels enabling us to quantify the accuracy of gradient descent in task abstraction space. We now make this connection in the introduction of the revised manuscript.
>
> Hummos, A. Thalamus: a brain-inspired algorithm for biologically-plausible continual learning and disentangled representations. (ICLR, 2023).
>
> Butz et al., Learning, planning, and control in a monolithic neural event inference architecture. Neural Networks 117, 135–144 (2019).
>
> Sandbrink et al., Neural networks with fast and bounded units learn flexible task abstractions. (NeurIPS 2024, spotlight)
>
> >The section on the one-step gradient update and maximal entropy initialisation could benefit from a clearer, more intuitive explanation. To improve clarity, the authors could add a visual schematic that illustrates the process step-by-step, and how a single gradient update shifts this initial state towards a more task-specific representation.
>
> Thank you for this thoughtful comment. We created the schematic and it does offer a more intuitive account of how gradients interact with the task abstraction space. This is currently in the new figure 1. Though we are still refining the plot. Thanks for this helpful comment.
>
> >Improving coherence: ..A bit more introduction and summarisation at the beginning and end of sections, focused on tying each section to the core ideas would improve the flow of the paper.
>
> Thank you for pointing out the need to better align our experiments with the paper’s motivations. We have made several changes to address this:
>
> 1. **Clarified the Benefits of GBI**: We now group the expected benefits into two categories: (i) during training—faster learning and reduced forgetting, and (ii) during testing—accurate task abstraction inference and recomposition for generalization.
> 2. **Updated Section Titles**: Section titles now explicitly state the key findings and align with the claims in the introduction.
> 3. **Added Introductions and Summaries**: Each experimental subsection begins with an overview of its goals tying the findings back to the paper’s central ideas.
>
> >limited use of intuitive examples: incorporating a few straightforward examples or analogies could provide readers with a more accessible understanding of the contributions being presented.
>
> Very valuable improvement to the paper. We now use an example inspired from Daniel Wolpert’s work on how motor feedback can be also seen as a mechanism to infer other people’s motives during social interactions. We added this to the introduction, and we summarize the example here:
>
> During upbringing, we may observe situations where people’s feelings were labeled as sadness or as anxiety. As we interact with others we may try to infer their emotional states relying on subtle cues. We incrementally adjust our conclusions with every cue, moving closer to one emotion or the other, as make predictions and receive feedback during the interaction. If, by the end of the interaction, both emotions seem equally likely, then either the situation is uncertain or that the person is experiencing an emotion outside of those two.

---

> > ### Author Response · Authors · 2024-11-25
> > **Response to v6Yp (2/2 comments)**
> >
> > >lack of comparison to important uncertainty estimation methods: while the paper claims advantages in OOD detection and uncertainty estimation, the results provided are not compared against established methods like ensemble approaches.
> >
> > We agree with the reviewer that our comparisons were rather limited. We now ran experiments for both ensemble models and Bayesian neural networks. While our previous manuscript compared only to Likelihood regret method as the method claimed state-of-the-art results, we found ensembles and BNNs to be surprisingly effective in the setting we study with normalized pixel intensities. (standard OOD tests leave the statistics of In and out of distribution datasets as is, which means models can trivially distinguish the two based on image brightness).
> >
> > We updated the table to include these comparisons. For reference here, ensemble methods had an AUROC of 0.809 $\pm$ 0.011, and BNNs  0.859 $\pm$ 0.040. Compared to our method at 0.89 $\pm$ 0.03. BNNs come close, but of course require 10 models trained, whereas our method trains only one.
> >
> > Of note, we did not run those on uncertainty estimation as this is not a claim for our method (GBI results on uncertainty are similar to a traditional classifier, with the small advantage that they do not require post hoc calibration). Those results are entirely in the Appendix.
> >
> > Thank you for this suggestion.
> >
> > >poorly organised code
> >
> > Thank you for your feedback on the code organization. To improve accessibility, we will restructure the repository with clear directories for models, experiments, and utilities, and add a detailed README.md with instructions for running experiments. We will also include configuration files for easy replication, enhance code readability with comments and docstrings, and provide a script to automate experiment execution. These updates will ensure the code is more user-friendly and easier to navigate.
> >
> > >given that the authors ground the problem setting and draw inspiration from human cognition and learning, do the authors feel their approach has biological plausibility at some level?
> >
> > An important topic for us indeed. We add a task abstraction layer to the model and we propose using gradients to infer the appropriate task abstractions. There are many ways to compute gradients, but our particular implementation relies on backpropagation through the neural network, which is not biologically plausible. However, we see at least two plausible solutions for an implementation in the brain. First, is a separate neural network that takes as inputs the output errors from the base model and outputs gradients for the task abstraction layer. Second, is node perturbation, whereby one can estimate the gradients by slightly moving the task abstraction values and observing the effects on the loss function at the output. Both of those are made incredibly more tractable because of the assumed low-dimensionality of the task abstraction layer, as opposed to using the same methods to estimate gradients for the high dimensional neural parameters.
> >
> > To summarize, we added 2 new schematics, tested ensembles and BNNs on OOD task, added an intuitive example for GBI, and improved the coherence of the manuscript. We hope these changes and the discussion here addressed the comments raised by the reviewer. We are grateful for the thoughtful improvements suggested and the explicit offer to revise the score.

---

> > > ### Comment · Reviewer_v6Yp · 2024-11-25
> > >
> > > Many thanks for your detailed responses to my and other reviewer's comments and concerns. I think the paper will benefit significantly from the substantial changes to the paper and code that you have proposed. Taking account of these plus the feedback from the other reviewers, I am raising my score from 3 to 5.

---

### Author Response · Authors · 2024-11-25
**General response to all reviewers**

We thank the reviewers for their thoughtful comments with ways to improve the presentation of the paper. We were relieved that the reviewers found the method “novel” (reviewers vL85, v6Yp), “innovative” (reviewer v6Yp), and “simple and effective” (reviewer MWGp). Reviewers also found the approach to have a “clear and relevant objective” (reviewer 4ta1) and “with a solid foundation.. in well-established principles” (v6Yp). Further, reviewer vL85 identified a new connection: the method might improve “interpretability” of neural networks.

In response to the reviewers’ comments, we made several key improvements to the manuscript:

1. **We now clearly state the goal of each experiment**: The results section has been restructured to clearly map each experiment to the benefits of task abstractions, including faster learning, reduced forgetting, and improved generalization and OOD detection. Figure 2 updated with two more panels to concretely show these benefits.

2. **New Schematic (Figure 1)**: We replaced the initial figure with a detailed schematic illustrating the framework, including what is optimized during training and inference.

3. **A new discuss on the motivation for using gradients for inference**: The introduction now explicitly motivates the use of gradients for task inference. Using gradients avoids misalignment between task abstractions and the network parameters they control during learning.

4. **Compare GBI to ensemble and Bayesian networks on OOD detection**: We ran additional experiments to compare OOD detection AUROC and implemented ensemble methods and Bayesian networks. GBI still shows better performance in comparison.

6. **Code Improvements and Additional Experiments**: We are actively organizing the code base, and also running further experiments on OOD detection on CIFAR 10. We will update this forum with progress on those fronts.

We hope our updates and discussions below address all comments raised and we are looking forward to participating in the discussion phase.

---

### Author Response · Authors · 2024-11-29

Dear Reviewers,

We appreciate your thoughtful feedback and the recognition that the paper introduces a simple and novel method to build neural networks that are responsive to task abstractions. It is clear that there is general agreement about the conceptual contributions of this work. However, there are differing perspectives on where the most impactful extensions of this work should lie.

Suggestions included testing the framework on other architectures (eg, transformer), scaling to larger image datasets, and analyzing how generalization occurs in the current smaller networks. While we understand the value of each of these directions, we focused our efforts on extending the framework in ways we felt were most conceptually significant and aligned with our goals. Specifically, we demonstrated the method on:

1.	A toy yet intuitive sequence prediction task, exposing a link to Bayesian inference.
2.	An image generation task with a larger number of task abstractions.
3.	A language modeling task, which we viewed as a critical milestone due to the challenges posed by discrete tokens, which raised a serious question about their compatibility with gradient descent continuous dynamics.

We acknowledge that there may be differences in opinion about the most valuable paths forward, and we respect the diverse perspectives brought to this discussion.

We conclude by highlighting what seems to be a shared recognition among reviewers: the method is simple, novel, and represents a conceptual advance in training neural networks to be responsive to task abstractions. Crucially, while such networks can depend on abstractions provided by humans or another network, our method also allows them to infer abstractions directly from data when needed.

Thank you again for your time, effort, and thoughtful engagement with our work.

Sincerely,

The Authors

---

### Meta-Review · Area_Chair_VC7K · 2024-12-21

**Metareview:**

This paper proposes to use Gradient-Based Inference (GBI) for learning abstract task representations, which allows neural networks to infer and adapt task representations dynamically. Inspired by human adaptability—where task abstractions allow flexible responses to the same input depending on internal goals—this approach enables neural networks to infer and adapt task representations on the fly. Experiments in a range of domains including image classification and language modelling demonstrate benefits in learning efficiency, generalisation, and reduced forgetting, as well as its performance in uncertainty estimation and out-of-distribution detection.

Strengths: The GBI method is novel and the problem it addresses is important. The connection to cognitive science and human cognition provides a solid grounding for this work.  The results show promising advantages in learning efficiency and generalization. The paper includes code which is appreciated and improves reproducibility.

Weaknesses:  The main concerns that remaining after the discussion relate to the experimental evaluation. The presented experiments are performed on simple datasets which serve as an intuitively appealing proof-of-concept, but do not provide strong empirical evidence for the advantages of the proposed method under realistic conditions and compared  to strong baselines.

This paper clearly has a lot of potential, but as it is, I consider it a borderline paper and lean towards rejection.

**Additional Comments On Reviewer Discussion:**

Authors and reviewers engaged in productive discussion that led to several important improvements to the paper, which is reflected in the improved score of reviewer v6Yp from 3 to 5.
The clarifications and additional experiments regarding ensemble and Bayesian network baselines definitely improve this paper and have pushed it from a clear rejection into borderline territory.

---

### Decision · Program_Chairs · 2025-01-22

Reject